



# Detection and Quantification of $CH_4$ Plumes using the WFM-DOAS retrieval on AVIRIS-NG hyperspectral data

Jakob Borchardt[1], Konstantin Gerilowski[1], Sven Krautwurst[1], Heinrich Bovensmann[1], Andrew K. Thorpe[2], David R. Thompson[2], Christian Frankenberg[3,2], Charles E. Miller[2], Riley M. Duren[4,2], and John Philip Burrows[1]

[1]Institute of Environmental Physics (IUP), University of Bremen, Bremen, Germany
[2]Jet Propulsion Laboratory, California Institute of Technology, Pasadena, CA, United States of America
[3]California Institute of Technology, Division of Geological and Planetary Sciences, Pasadena, CA, United States of America
[4]Institutes for Resilience, University of Arizona, Tuscon, AZ, United States of America

**Correspondence:** Jakob Borchardt (jakob.borchardt@iup.physik.uni-bremen.de)

**Abstract.** Methane is the second most important anthropogenic greenhouse gas in the Earth's atmosphere. Reducing methane emissions is consequently an important element in limiting the global temperature increase below 2 °C compared to preindustrial times. Therefore, a good knowledge of source strengths and source locations is required. Anthropogenic methane emissions often originate from point sources or small areal sources, such as fugitive emissions at oil and gas production sites or landfills. Airborne remote sensing instruments such as the Airborne Visible InfraRed Imaging Spectrometer - Next Generation (AVIRIS-NG) with meter scale imaging capabilities are able to yield information about the locations and magnitudes of methane sources, especially in areas with many potential emission sources.

To extract methane column enhancement information from spectra recorded with the AVIRIS-NG instrument, different retrieval algorithms have been used, e.g. the matched filter (MF) or the Iterative Maximum A Posteriori DOAS (IMAP-DOAS) retrieval. The WFM-DOAS algorithm, successfully applied to AVIRIS-NG data in this study, fills a gap between those retrieval approaches by being a fast, non-iterative algorithm based on a first order approximation of the Lambert-Beer law, which calculates the change in gas concentrations from deviations from one background radiative transfer calculation using precalculated weighting functions specific to the state of the atmosphere during the overflight. This allows the fast quantitative processing of large data sets. Although developed for high spectral resolution measurements from satellite instruments such as SCIAMACHY, TROPOMI and the MAMAP airborne sensor, the algorithm can be applied well to lower spectral resolution AVIRIS-NG measurements. The data set examined here was recorded in Canada over different gas and coal extraction sites as part of the larger Arctic Boreal Vulnerability Experiment (ABoVE) Airborne Campaign in 2017.

The noise of the retrieved $CH_4$ imagery over bright surfaces ($> 1\,\mu W\,cm^{-2}\,nm^{-1}\,sr^{-1}$ at $2140\,nm$) was typically $\pm 2.3\%$ of the background total column of $CH_4$ when fitting strong absorption lines around $2300\,nm$, but could reach over $\pm 5\%$ for darker surfaces ($< 0.3\,\mu W\,cm^{-2}\,nm^{-1}\,sr^{-1}$ at $2140\,nm$). Additionally, a worst case large scale bias due to the assumptions made in the WFM-DOAS retrieval was estimated to be $\pm 5.4\%$. Radiance and fit quality filters were implemented to exclude the most uncertain results from further analysis, mostly due to either dark surfaces or surfaces, where the surface spectral reflection structures are similar to $CH_4$ absorption features at the spectral resolution of the AVIRIS-NG instrument.





We detected several methane plumes in the AVIRIS-NG images recorded during the ABoVE Airborne Campaign. For four of those plumes, the emissions were estimated using a simple cross sectional flux method. The retrieved fluxes originated from well pads and cold vents and ranged between $(89 \pm 46)\,\mathrm{kg}\,(\mathrm{CH}_4)\,\mathrm{h}^{-1}$ and $(141 \pm 87)\,\mathrm{kg}\,(\mathrm{CH}_4)\,\mathrm{h}^{-1}$. The wind uncertainty was a significant source of uncertainty for all plumes, followed by the single pixel retrieval noise and the uncertainty due to

atmospheric variability. For one plume the wind was too low to estimate a trustworthy emission rate, although a plume was visible.

## 1 Introduction

Methane ($\mathrm{CH}_4$) is an important greenhouse gas, with a global warming potential approximately 28 times larger than that of carbon dioxide ($\mathrm{CO}_2$) on a timescale of 100 years (IPCC 2013, 2013). After a brief period of stable mixing ratios at the

beginning of the 21st century, $\mathrm{CH}_4$ concentrations have again begun rising (Dlugokencky et al., 2011; Dlugokencky, 2018), though the origins of this stabilization and renewed increase are still debated (see for example Schaefer, 2019, and references therein). This uncertainty emphasizes the need to reduce anthropogenic $\mathrm{CH}_4$ emissions to reach the goal of the Paris Agreement (Paris Agreement, 2015; Nisbet et al., 2019).

According to the Global Carbon Project (GCP, Saunois et al., 2016, 2019) between $\sim 50\,\%$ and $\sim 60\,\%$ of the global methane

emissions are anthropogenic. Of those, roughly $55\,\%$ result from agricultural practices and waste management, and nearly $35\,\%$ from losses during fossil fuel extraction, delivery and use in energy production and transport, with a small contribution ($\sim 10\,\%$) coming from biomass and biofuel burning. Satellite instruments such as SCIAMACHY (spatial resolution $\sim 30 \times 60\,\mathrm{km}^2$, Burrows et al., 1995; Bovensmann et al., 1999) and TROPOMI (spatial resolution $\sim 7 \times 7\,\mathrm{km}^2$, Veefkind et al., 2012; Hu et al., 2016) have successfully been used to assess methane emissions from emission hot spots (Frankenberg et al., 2006; Schneising

et al., 2009; Buchwitz et al., 2017; Hu et al., 2018; Schneising et al., 2019; Pandey et al., 2019). Additional efforts to make single $\mathrm{CH}_4$ emitters visible from space by using measurements with high spatial resolution have been made, but so far only strong single sources have been quantified (Thompson et al., 2016; Varon et al., 2019). Many anthropogenic $\mathrm{CH}_4$ emissions occur over relatively large areas (e.g. rice paddies, animal herds, landfills), at previously unknown point sources (e.g. pipeline leaks, broken valves) or with highly varying emissions, making reliable detection and attribution of single sources from space

challenging.

Airborne remote sensing campaigns can often gain a better knowledge of single emitter source strengths in emission hot spot regions due to their higher spatial resolution. In these campaigns, a defined area is sampled with stronger sensitivity to small localized $\mathrm{CH}_4$ sources. For example, the Methane Airborne MAPper (MAMAP, Gerilowski et al., 2011), a non-imaging instrument with a nadir pointing field of view and a high spectral resolution of $\sim 0.9\,\mathrm{nm}$, successfully quantified emissions

of known sources like coal mining shafts (Krings et al., 2013) or smaller areal sources like landfills (Krautwurst et al., 2017). However, its viewing geometry requires a flight pattern orthogonal to the $\mathrm{CH}_4$ plume for emission estimates and therefore limits the potential to pinpoint single unknown sources in a field of multiple potential sources. This problem is solved by using airborne imaging systems, which take multiple measurements across the flight track, thus creating an image of the area they





pass over. However, to our knowledge, there is not yet an operational airborne imaging instrument specifically designed and optimized for the detection of $CH_4$ available.

Nevertheless, data from multiple imaging instruments have been analyzed to map and/or quantify $CH_4$ emissions. For example, thermal imaging instruments such as SEBASS (Spatially-Enhanced Broadband Array Spectrograph System) could detect
methane plumes as low as $0.4\,kg\,h^{-1}$ of $CH_4$ flying $500-700\,m$ above ground during a controlled release experiment (Scafutto et al., 2018). Also, the HyTES (Hyperspectral Thermal Emission Spectrometer) instrument has demonstrated detection of $CH_4$ leaks (Hulley et al., 2016). While these instruments have been able to detect very small sources at low flight altitude ($500\,m$ above ground), performance may suffer at higher altitudes. For example, HyTES flying at $3\,km$ had some difficulty consistently detecting a coal mine ventilation shaft plume with an estimated emission of $\sim 1650\,kg\,h^{-1}$ (Jongaramrungruang et al., 2019)
due to the strongly varying sensitivity of the instrument to different atmospheric layers.

In the shortwave infrared (SWIR), the Airborne Visible-Infrared Imaging Spectrometer - Next Generation (AVIRIS-NG) was used for the detection and quantification of anthropogenic methane sources (Thompson et al., 2015; Frankenberg et al., 2016; Thorpe et al., 2016a, 2017; Duren et al., 2019; Cusworth et al., 2020; Thorpe et al., 2020). As the instrument was not designed for the detection of atmospheric absorbers, it has a spectral resolution much coarser than SWIR instruments
specifically designed to measure $CO_2$ and $CH_4$. However, AVIRIS-NG has a very high signal to noise ratio (SNR) and meter scale spatial resolution. The latter depends on flight altitude and flight speed, with typical values for the ground sampling distances for large scale methane surveys of $3 \times 3\,m^2$ to $5 \times 5\,m^2$. Successful algorithms for the retrieval of methane comprised either a matched filter approach (Thompson et al., 2015), which uses a hypothesis test between presence and absence of additional $CH_4$ to infer $CH_4$ increases, or an adaption of the IMAP-DOAS retrieval (Frankenberg et al., 2005) to AVIRIS-
NG airborne data (Thorpe et al., 2013, 2017), which is an iterative optimal estimation based algorithm. However, the latter is computationally very expensive, which makes it less suited to analyze large data sets aquired during longer measurement campaigns (Thorpe et al., 2017). Consequently, it has only been applied to regions of special interest in the data.

In this study, we test and apply an adaption of the WFM-DOAS algorithm, used previously for the higher spectral resolution MAMAP measurements (spectral resolution $\sim 0.9\,nm$ Krings et al., 2011), to hyperspectral AVIRIS-NG data (spectral
resolution $\sim 6\,nm$). The data set was aquired during the Arctic Boreal Vulnerability Experiment Airborne Campaign (ABoVE, Miller et al., 2019b) in Canada and Alaska, which included overflights of multiple coal, oil and gas production sites. The WFM-DOAS approach uses assumptions on the background state of the atmosphere at the time and location of the overflight, including scattering. It performs a linear fit of atmospheric parameters deviating from this background state, making it a fast quantitative method compared to iterative retrievals. We identified multiple plumes in the retrieval results, and for four of them,
the emissions were estimated by application of a cross sectional flux method.

This publication is organized as follows: Following this introduction, Sect. 2 gives an overview of the instrument and data sets. Section 2.1 describes the AVIRIS-NG instrument and radiance data, and Sect. 2.2 introduces the ERA-5 meteorological data briefly. In Sect. 3, we present the retrieval algorithm for $CH_4$ and the subsequent filtering. First, we describe the WFM-DOAS method used to infer methane enhancement maps from the spectra in Sect. 3.1. Section 3.2 justifies the fitting windows
we use in the retrieval. Sect. 3.3 evaluates the sensitivity of the retrieval to assumptions in the forward model and in Sect.





3.4 we implement a filtering to remove certain error cases. We present experimental results in Sect. 4. First, the detection of plumes is described in Sect. 4.1. Second, Sect. 4.2 illustrates a flux inversion using the cross sectional flux method, and Sect. 4.3 finally shows the results and uncertainties of the emission estimate for four plumes. The results are discussed in Sect. 5. Sect. 6 summarizes the findings of this study.

## 2 Instrument and data sets

### 2.1 The AVIRIS-NG instrument and measurements

AVIRIS-NG is a hyperspectral imaging spectrometer with a spectral sampling of $\sim 5\,\mathrm{nm}$ and a spectral resolution of $\sim 5-6\,\mathrm{nm}$, depending on the wavelength (Hamlin et al., 2011). As a nadir looking instrument, it measures solar radiation reflected from ground in the wavelength range from $380$ to $2450\,\mathrm{nm}$ with a high signal-to-noise ratio of up to $800$ at $2200\,\mathrm{nm}$ (Thorpe et al., 2016a). The instrument contains 600 spatial pixels, each having a 1 mrad field of view. This results in individual samples with 5 m spatial resolution and a 3 km swath from a typical flight altitude of 5 km above ground level. This allows it to scan large areas in short periods of time. The level-1 data distributed by the operations team contains orthorectified (and gridded) absolute radiances (Chapman et al., 2019), with additional data containing observation parameters such as flight altitude, both solar and instrument zenith and azimuth angles and surface elevation among others (see Miller et al., 2019a, for data description).

For this study, we analyzed a subset of the measurements collected during the ABoVE Airborne Campaign (Miller et al., 2019b) in 2017. The ABoVE campaign aimed to better understand the impacts of environmental changes in Alaska and western Canada. During the airborne campaign, several flight lines of the AVIRIS-NG instrument covered fossil fuel infrastructure in Canada, which contained multiple potential sources for $CH_4$ emission plumes.

The data analyzed in this paper had been preselected to cover a wide range of surface types (e.g. forest, mountainous regions, sand, grass). Additionally, the tracks contained different emission sources detected using the matched filter (MF) algorithm (Thompson et al., 2015) to test the retrieval algorithm against known plumes over different terrain. The preselection contained 13 flight lines on 5 different days in August 2017, covering different types of sources and surface types.

### 2.2 ERA-5 meteorological data

The WFM-DOAS retrieval and the flux inversion require information about various atmospheric parameters in addition to the observed radiances. The following meteorological parameters were extracted from ERA-5 reanalysis data (Copernikus Climate Change Service (C3S) (2017), 2017): hourly data of temperature, pressure and water vapor profiles, as well as height resolved wind speeds and wind components.

For a given flight line, the atmospheric parameters of the nearest 4 spatial grid points and the nearest two time steps of the ERA-5 data set were interpolated to the time and location of the flight line. For the wind speed and the wind direction, the wind speed components in East and North directions were averaged over the lowest $50\,\mathrm{m}$. We used the averaged wind speed for the





inversion, and included the averaged wind direction in the plots of the detected plumes. The wind direction for the inversion was estimated from the plume structure itself.

## 3 Adaption of WFM-DOAS algorithm to AVIRIS-NG measurements

The WFM-DOAS algorithm was first developed for SCIAMACHY measurements (Buchwitz et al., 2000; Schneising et al.,
2008), where the absorption bands around $1580\,\mathrm{nm}$ and $1660\,\mathrm{nm}$ were used for the retrieval of $CO_2$ and $CH_4$. Recently, it has been modified and applied to TROPOMI measurements by Schneising et al. (2019) for the simultaneous retrieval of $CH_4$ and CO. As TROPOMI was build without spectral bands around $1600\,\mathrm{nm}$, the retrieval used the wavelength ranges from $2311\,\mathrm{nm} - 2315.5\,\mathrm{nm}$ for CO and from $2320\,\mathrm{nm} - 2338\,\mathrm{nm}$ for $CH_4$.

Additionally, the WFM-DOAS algorithm has been adapted and used since 2007 to retrieve local $CH_4$ and $CO_2$ enhancements
from MAMAP aircraft measurements in the wavelength range between $1590\,\mathrm{nm}$ and $1690\,\mathrm{nm}$ (see e.g. Krings et al., 2011, 2013; Krautwurst et al., 2017; Krings et al., 2018).

### 3.1 Retrieval of total column increases with WFM-DOAS

The WFM-DOAS algorithm minimizes the difference between a measured and a modeled spectrum by scaling weighting functions for the different trace gas profiles such as $CH_4$ and $CO_2$, shifting the temperature profile, and fitting a low order
polynomial for broad band absorption (e.g. at the surface) or scattering (e.g. by air molecules and aerosols). The weighting functions represent a linear relationship between the change in observed radiance and a change in the atmospheric parameters. A detailed mathematical description of the WFM-DOAS algorithm modified for aircraft measurements and an analysis of this method for MAMAP measurements can be found in Krings et al. (2011).

Each flight track covered a different scene or different day. We calculated a modeled spectrum for each scene using the SCI-
ATRAN radiative transfer model (Rozanov et al., 2017). There we used the solar zenith angle, viewing angle of the instrument, solar and instrument azimuth angle, surface elevation and flight altitude from the AVIRIS-NG level 1 orthorectified data set (see data set description in Miller et al., 2019a). For each flight track, we calculated the mean value of each parameter and used the results as input for the radiative transfer calculation.

For the radiative transfer calculation with SCIATRAN, the state of the atmosphere for the location of the flight track during
the time of overflight was equally important. Temperature, pressure and water vapor profiles were extracted from ECMWF ERA5 meteorological data (see 2.2). The background total columns of carbon dioxide ($CO_{2,\mathrm{back}}$) were calculated using the Simple Empirical $CO_2$ Model (SECM) by Reuter et al. (2012) in the version SECM2018, which contains a recently updated parameter set (see also Reuter et al., 2020). The background total columns of methane ($CH_{4,\mathrm{back}}$) were calculated with the approach used by Schneising et al. (2019), where a climatology averaged over the years 2003 - 2005 was enhanced by the total
increase in methane based on globally averaged marine NOAA surface data (Dlugokencky, 2018). The $CH_4$ and $CO_2$ profiles used in SCIATRAN where then obtained by scaling a US Standard Atmosphere (Oceanic et al., 1976) so that the total column mixing ratio calculated from those profiles matched the a priori estimated local total column mixing ratio. HITRAN 2016

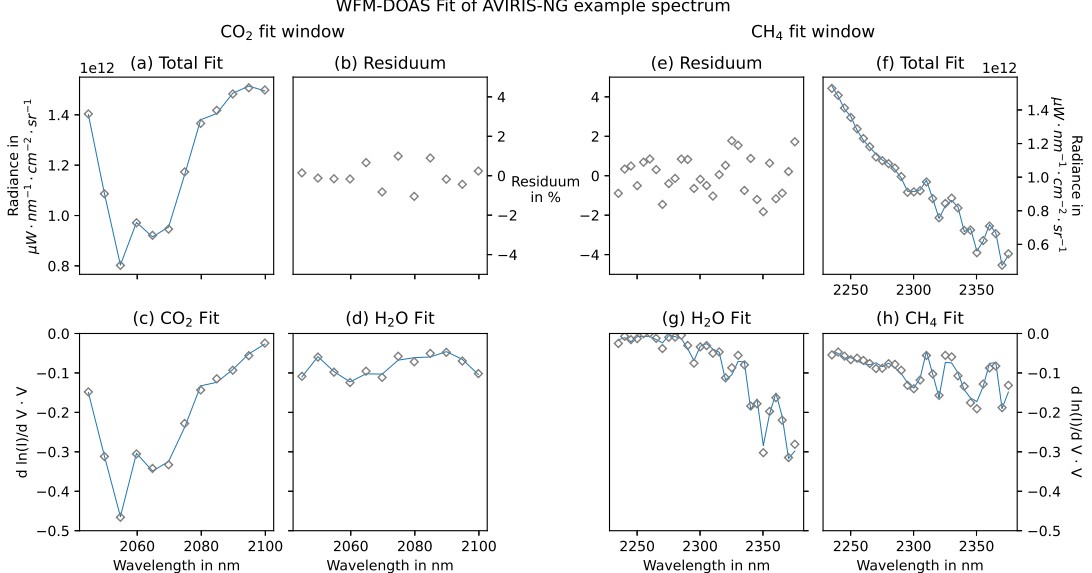

**Figure 1.** Example fit result of the WFM-DOAS retrieval for AVIRIS-NG data. The left block ((a) - (d)) shows the $CO_2$ fitting window ($2040\,\text{nm} - 2100\,\text{nm}$), the right block ((e) - (h)) shows the $CH_4$ fitting window ($2235\,\text{nm} - 2380\,\text{nm}$). In sub figures (a) and (h) the fit result is shown as a solid blue line, while the actual measured intensities are shown as gray diamonds. The difference between the gray diamonds and the line is the residuum, which is shown in sub figures (b) and (e). In the second row, the scaled weighting functions for $CO_2$ and $H_2O$ in the $CO_2$ window ((c) and (d)) and for $CH_4$ and $H_2O$ in the $CH_4$ window ((g) and (h)) are shown as solid lines. The gray diamonds show the fit result plus the residuum, to assess, if the residual structures are larger than the actual fitted structures. In the $CH_4$ fit window the residuum shows some structure, which might indicate some residual correlation between water vapor and methane signature in the real measurements, which can not fully be resolved by the retrieval.

(Gordon et al., 2017) was used as spectral line parameter data base in SCIATRAN for trace gas absorption. The SCIATRAN model predicted the radiance at the sensor for the background case and the height dependent weighting functions for $CH_4$, $CO_2$ and $H_2O$.

The WFM-DOAS retrieval produced profile scaling factors (PSF), which scaled the weighting functions of $CH_4$, $CO_2$, $H_2O$

5 and temperature in a linear fit. An example fit of AVIRIS-NG data with the resulting residual structures is shown in Fig. 1. The light passed the air mass above the aircraft once on the downward path to the Earth, but transected the air mass below the aircraft on both downward and upward paths. Consequently, the retrieval was more sensitive to atmospheric changes below the aircraft than above. This was captured by the averaging kernel, which represented the sensitivity of the instrument to changes in a specific altitude layer. In our case, strong local enhancements in atmospheric methane were confined below the aircraft,

10 so we multiplied the total column enhancements by the inverse of the averaging kernel for the airmass underneath the aircraft ($k_{\text{AK}}$) to determine the true enhancement of $CH_4$ caused by an emission source near the ground.



We did not retrieve the pressure profile or scattering properties. This, and also other effects like surface elevation changes could alter the light path and therefore the absorption strength, which would be detected as an enhancement. We used the proxy method to correct the retrieved column enhancement for those effects (see also Frankenberg et al., 2005; Schneising et al., 2009; Krings et al., 2011). Specifically, we divided the derived scaling factor of $CH_4$ by the scaling factor of another well

mixed gas, which was assumed to be constant for the region of interest and time of overflight. For this work, we used $CO_2$ as a proxy because of its spectral proximity to the $CH_4$ absorption band, resulting in the proxy

$$PSF_{CH_4, proxy} = PSF_{CH_4}/PSF_{CO_2}. \tag{1}$$

Finally, we corrected the enhancements in a detected plume for large scale effects by normalizing over the local background around the plume ($PSF_{proxy, bg}$). The local column enhancement of $CH_4$ below the aircraft in a plume ($CH_{4,enh}$) was then

$$CH_{4,enh} = \left( \frac{PSF_{CH_4}}{PSF_{CO_2}} \Big/ PSF_{proxy, bg} - 1 \right) \cdot CH_{4,back} \cdot k_{AK}. \tag{2}$$

A discussion of the biases introduced by the assumptions made for the WFM-DOAS retrieval are studied in the sensitivity analysis in Sect. 3.3.

### 3.2   Comparison of major fitting windows in the SWIR spectral range

For the MAMAP instrument the fitting window was $\sim 1630\,nm - 1675\,nm$ for $CH_4$ and $\sim 1592\,nm - 1617\,nm$ for $CO_2$ at

$0.9\,nm$ spectral resolution due to the sensor design (see Krings et al., 2011; Gerilowski et al., 2011). AVIRIS-NG additionally offered the possibility to fit the $CH_4$ and $CO_2$ absorption lines between $2000\,nm$ and $2400\,nm$ for the retrieval of $CH_4$, although at a coarser spectral resolution of $5.5\,nm - 6.0\,nm$. For example, the IMAP-DOAS retrieval successfully retrieved $CH_4$ concentrations from AVIRIS-NG data using the spectral regions of $2215\,nm - 2410\,nm$ for $CH_4$ and $1904\,nm - 2099\,nm$ for $CO_2$ (Thorpe et al., 2017).

For a first assessment of those absorption bands, we convolved a simulated high resolution spectrum and the corresponding weighting functions for $CH_4$, $CO_2$ and $H_2O$ with the AVIRIS-NG instrument spectral response function. We used a Gaussian spectral response, where the FWHM were distributed as part of the data set. Fig. 2 shows the results of convolution and resampling to the AVIRIS-NG wavelength grid.

Both fitting windows had their advantages and disadvantages, especially for the lower spectral resolution of AVIRIS-NG.

Around $2300\,nm$, the absorption features of $CH_4$ were about a factor of 2 stronger and had a more pronounced structure, which could lead to a better detection of methane changes. Around $1650\,nm$, the at sensor radiance was nearly twice as high for the same albedo, which could mean a higher signal-to-noise ratio. Additionally, there was less overlap with water vapor absorption features near $1600\,nm$.

We used a two step approach to find the best fitting window: First, we created a spatially averaged spectrum over a ho-

mogeneous surface elevation and surface type to reduce the instrument noise and systematic influences. Then, we optimized the edges of both fit windows for fitting the gas features in each window. As a measure of fit quality, the root mean square error (RMSE) between measurement and fit result was used. For $CH_4$, the best fitting windows were $1625\,nm - 1700\,nm$ and





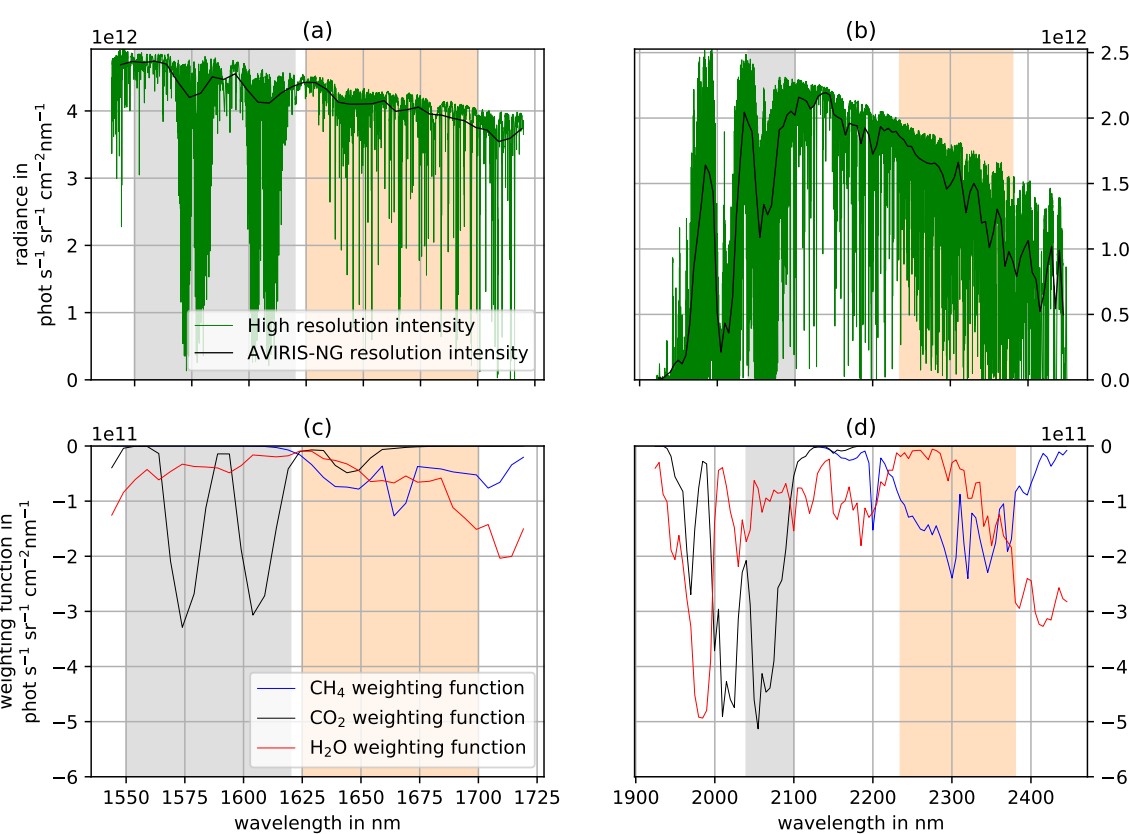

**Figure 2.** The high resolution simulated spectra (in (a) and (b) green) are convolved with the slit function of AVIRIS-NG and sampled to the AVIRIS-NG wavelength grid (solid black line in upper row). The lower row ((c) and (d)) shows the weighting functions, i.e. the change of intensity due to a change in atmospheric concentration for $CH_4$ (blue), $CO_2$ (black) and $H_2O$ (red). The shaded areas denote the fitting windows for $CO_2$ (gray) and $CH_4$ (light orange). These have been chosen to be large enough to have enough data points inside the fitting window, without interfering with other gases (Sect. 3.2).

$2235\,nm - 2380\,nm$, and for $CO_2$ $1550\,nm - 1620\,nm$ and $2040\,nm - 2100\,nm$ (see also Fig. 2). For simplicity, the fitting windows between $1550\,nm$ and $1700\,nm$ will be called "weak windows", and the fitting windows between $2040\,nm$ and $2380\,nm$ will be called "strong windows" in the following parts, according to the depth of the absorption features.

5    To assess the measurement precision in each window, we selected a homogeneous, flat, bright area which contained no potential sources. We then applied the retrieval to the whole flight line containing this test case for each of the fitting windows and gases. These initial results showed detector column dependent stripes (see Sect. A in the appendix). To correct this effect,





**Table 1.** Comparison of the standard deviation of $PSF_{CH_4}$, $PSF_{CO_4}$ and $PSF_{CH_4,proxy}$ in the two fitting windows around $1645\,nm$ and $2300\,nm$ for the AVIRIS-NG FWHM ($\approx 6\,nm$). The standard deviation was calculated over a homogeneous and flat area with no visible plume and possible source inside. The statistical uncertainties for $PSF_{CH_4}$ and $PSF_{CO_2}$ are therefore uncorrelated.

|  | Standard deviation PSF 1645 nm fitting window | Standard deviation PSF 2300 nm fitting window |
|---|---|---|
| $PSF_{CH_4}$ | $\pm 6.4\%$ | $\pm 1.9\%$ |
| $PSF_{CO_2}$ | $\pm 1.9\%$ | $\pm 1.3\%$ |
| $PSF_{CH_4,proxy}$ | $\pm 6.6\%$ | $\pm 2.3\%$ |

we normalized the $PSF_{CH_4}$, $PSF_{CO_2}$ and $PSF_{CH_4,proxy}$ for each pixel by the median PSF of its corresponding detector column. We selected the median for resilience against outliers, which could otherwise have a large impact on the correction

After destriping, we compared the standard deviation in the test case region of the weak and strong window retrieval results of $PSF_{CH_4}$, $PSF_{CO_2}$ and $PSF_{CH_4,proxy}$ (Table 1. The retrieved $PSF_{CH_4}$ and $PSF_{CH_4,proxy}$ were noisier in the weak window by a
factor of 3.3 and 2.9, respectively. The retrieved $PSF_{CO_2}$ was noisier by a factor of 1.5 in the weak window. Therefore, we only used the strong windows in later analyses.

### 3.3 Sensitivity analysis

In addition to the noise in the spectra, uncertainties and variability in the assumed constant atmospheric background parameters could lead to errors in the retrieval results. To assess the magnitude and influence of these deviations, we performed multiple
sensitivity analyses. We used a common set of geometric and atmospheric parameters to model the background spectrum. We then perturbed these atmospheric parameters to create synthetic AVIRIS-NG observations at instrument spectral resolution. Next, we applied the WFM-DOAS algorithm to these simulated measurements, and assessed the systematic offset from the expected PSF value for $PSF_{CH_4}$, $PSF_{CO_2}$ and $PSF_{CH_4,proxy}$. To assess the influence of linearization on the retrieval results, we did not include instrument noise in this analysis. The background simulation was based on the parameters extracted for one
flight line observed with a nadir viewing angle. The CH$_4$ enhancements and a plume from this flight line are shown in Fig. 8.

For the sensitivity analysis, we perturbed the following set of parameters (Table 2): the aircraft altitude, the surface elevation, the instrument viewing angle (i.e. the instrument zenith angle) and the surface albedo as geometric parameters; and the total columns of $CH_4$, $CO_2$ and $H_2O$, and the pressure and temperature profiles as atmospheric parameters. Additionally, we used selected spectral reflectance spectra of different surfaces instead of a spectrally uniform albedo and examined two additional
aerosol scenarios. We did not analyze the sensitivity to the solar zenith and azimuth angles, since these angles were effectively constant over the timespan of a flight line. In addition, we did not analyze the instrument azimuth angle dependency since the flight tracks were nearly straight and the azimuth angle therefore effectively constant for a flight line.

The viewing angle variations were chosen to represent the range of the AVIRIS-NG viewing angles. The surface elevation and aircraft altitude deviation were chosen to represent plausible deviations over one flight line. Temperature deviations were



chosen to be relatively large, as the temperature profile at the time of overflight for the specific ground scene might deviate quite a lot from the ERA-5 reanalysis due to the spatial and temporal resolution of the model output. The pressure scaling was chosen to represent a possible range of deviations, erring in favor of a conservatively high deviation for the observed scales. The albedo deviations covered the range which was expected around $2100\,nm - 2300\,nm$ (Chen et al., 2006).

The scaling of the $CO_2$ column and the $H_2O$ column spanned the natural deviations of $CO_2$ and water vapor from the assumed background to establish an upper bound on errors from these effects. The range for the total column of $CH_4$ covered the range, which might be observed directly over or near a strong source. However, in most ground scenes containing a plume signal, the enhancement was well below $20\,\%$, and for smaller plumes normally even near the source well below $10\,\%$.

    The reflectance spectra in the sensitivity analysis included surfaces present in the survey region or associated with oil and
gas infrastructure. The spectral reflectances were based on the ECOSTRESS Spectral library (Meerdink et al., 2019; Baldridge et al., 2009) and on the US Geological Survey Spectral Library, Version 7 (Kokaly et al., 2017). They contained spectra from a surface covered by a typical plant of the Canadian savanna, sandstone, sand, and rangeland surfaces, and anthropogenic structures such as aluminium, steel and paving substances. The reflectance spectra are shown in Fig. 3.

    The background aerosol scenario was assumed to be an OPAC (Hess et al., 1998) urban aerosol scenario (same as used
in Krings et al., 2011), as we were interested in emissions from anthropogenic infrastructure. To determine the magnitude of influence of the aerosol scenario on the retrieval, we used additionally simulated measurements with an OPAC background and desert aerosol scenario.

    After retrieving the profile scaling factors of $CH_4$ and $CO_2$ for each simulation, we calculated their deviation from the ground truth defined in the simulations. We also calculated the deviations for the $CH_4$ proxy method described in Sect. 3.1, and plotted
the errors as a function of the perturbation of each parameter in Fig. 4. While the observed uncertainties for the single profile scaling factors were quite high (orange and blue curves), for example up to $10\,\%$ for an elevation change of approximately $400\,m$, they were highly reduced by the proxy method (green curve). The influence of the surface spectral reflectance is shown in Table 3 and discussed at the end of the section.

    For aircraft altitude, temperature shifts, surface pressure scaling and viewing angle of the instrument, the maximum deviation
from the ground truth for $PSF_{CH_4,proxy}$ remained well below $0.5\,\%$. Also for most albedo values the maximum deviation remained below $0.5\,\%$. For a very low albedo of 0.01, results of the single retrievals as well as the proxy method both degraded considerably. We examined low radiance ground scenes further in Sect. 3.4. For large perturbations of the surface elevation of $400\,m$, the proxy method reduced the error only to around $\pm 3.6\,\%$. The different aerosol scenarios did not introduce major errors either. For the OPAC background scenario, the error was well below $0.1\,\%$, and even for the OPAC desert aerosol load,
only an error of $-0.2\,\%$ was introduced.

    For perturbations of $CH_4$ and $H_2O$, errors between the true and retrieved PSF growed, the larger the perturbations got. WFM-DOAS assumes a linear relationship between gas enhancement and radiance, but this assumption does not hold for large deviations from the background. This will also occur for $CO_2$ when choosing larger deviations from the background.

    When we perturbed $CO_2$, the application of the proxy method increased the error in methane. When only $CO_2$ was varied,
the methane column alone was retrieved correctly in the standard retrieval. Similarly, the retrieval correctly estimated the total



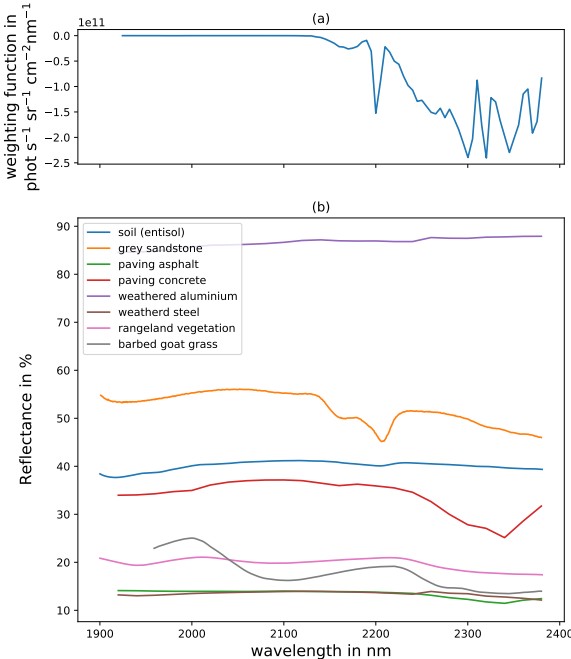

**Figure 3.** In (a), the weighting function for $CH_4$ is displayed. In (b), the reflectance spectra covered in the sensitivity analysis are shown. Especially for the paving concrete one can see a similar broad band shape compared to the weighting function of $CH_4$, which is caused by calcium carbonate (limestone).

column of $CO_2$. However, in the proxy method the retrieved $PSF_{CH_4}$ was divided by the retrieved $PSF_{CO_2}$, so that a decrease in $CO_2$ led to an apparent increase in $CH_4$ and vice versa. This meant, that $CO_2$ emission sources could mask $CH_4$ emissions, if the relative single column enhancement of $CO_2$ is similar or greater than that of $CH_4$. As the retrieval noise is similar for both gases (Table 1), this would be visible as a $CO_2$ point source in the $PSF_{CO_2}$ map.

5  For the scaling of $CH_4$, the proxy method did not reduce the deviation of the retrieved from the true enhancements, as expected. However, the large deviations for strong enhancements ($11\%$ underestimation for $100\%$ increase) would nevertheless mean a clearly detectable signal in the retrieved $CH_4$ maps. Smaller deviations ($\pm 20\%$) from the background profile would induce only small ($< 1\%$) underestimations. Consequently, for inversions of large emitters, the emission might be underestimated near the source, where the large enhancements are located. In cases with large concentrations near the source, emission

10  estimates should only be performed further down the plume.

To estimate the total systematic uncertainty, we combined all uncertainties in Table 4 aside from the extreme $CH_4$ case in quadrature. This led to maximum systematic uncertainties of $\pm 8.0\%$ for $PSF_{CH_4}$, $\pm 11.3\%$ for $PSF_{CO_2}$, and $\pm 5.4\%$ for





**Table 2.** Parameters studied in the sensitivity analysis and the range in which deviations were analyzed. The second column shows the background scenario used as "truth" in the sensitivity study. The third column notes the range of the perturbation of the parameters. Parameters not mentioned here were constant and estimated as described in Sect. 3.1 for the flight line ang20170811t192639.

| Parameter | Standard value | Studied range |
|---|---|---|
| Aircraft altitude | $5.33\,\mathrm{km}$ | $4.93\,\mathrm{km}$ to $5.93\,\mathrm{km}$ |
| Surface elevation | $0.39\,\mathrm{km}$ | $0.0\,\mathrm{km}$ to $0.6\,\mathrm{km}$ |
| Instrument viewing angle | $0.00\,°$ | $\pm 18\,°$ |
| Surface albedo | $0.1$ | $0.01$ to $0.5$ |
| $xCH_4$ | $1.833\,\mathrm{ppm}$ | $(0.5\ \mathrm{to}\ 2)\cdot 1.833\,\mathrm{ppm}$ |
| $xCO_2$ | $399.2\,\mathrm{ppm}$ | $(0.97\ \mathrm{to}\ 1.03)\cdot 399.2\,\mathrm{ppm}$ |
| $H_2O$ | $5.94\cdot 10^{22}\,\mathrm{molec\,cm^{-2}}$ | $(0.5\ \mathrm{to}\ 2)\cdot 5.94\cdot 10^{22}\,\mathrm{molec\,cm^{-2}}$ |
| Pressure profile | US standard scaled to $1015\,\mathrm{hPa}$ at sea level | $(0.95\ \mathrm{to}\ 1.05)\cdot 1015\,\mathrm{hPa}$ |
| Temperature profile | US standard shifted to $299\,\mathrm{K}$ at surface | $\pm 10\,\mathrm{K}$ |
| Aerosol scenario | OPAC urban | OPAC background, OPAC desert |
| Surface reflectance | Constant albedo $0.1$ | Rangeland vegetation<br>Barbed goatgrass<br>Soil (Entisol)<br>Grey sandstone<br>Weathered aluminium<br>Weathered steel<br>Paving asphalt<br>Paving concrete |

$PSF_{\mathrm{CH_4,proxy}}$ due to the simplification of the radiative transfer calculation to one single background spectrum and set of weighting functions. This uncertainty defined the large scale deviations possible in one flight track and should not be confused with the single pixel precision of the column enhancement, nor did that automatically limit detection. As parameters such as surface elevation in normal cases only vary smoothly, a plume signal on top of this bias is still detectable. However, a problem may

5 occur, if large amounts of $CO_2$ are co-emitted with a weak methane source. In this case there could be a (partial) masking of the plume due to the negative bias introduced by $CO_2$.

In contrast to those biases, the different surface types induce widely varying biases (Table 3) at AVIRIS-NG spectral resolution. The proxy method reduced these errors for some surfaces, but not all. For rangeland vegetation, soil, gray sandstone and weathered aluminium the bias after application of the proxy was well below $1\,\%$. However, for weathered steel and paving

10 asphalt, the bias increased to $1-2\,\%$, while for barbed goat grass and paving concrete (i.e. limestone) the bias due to the reflection properties was greater than $7\,\%$. This meant, that even after application of the proxy, some residual influence of the

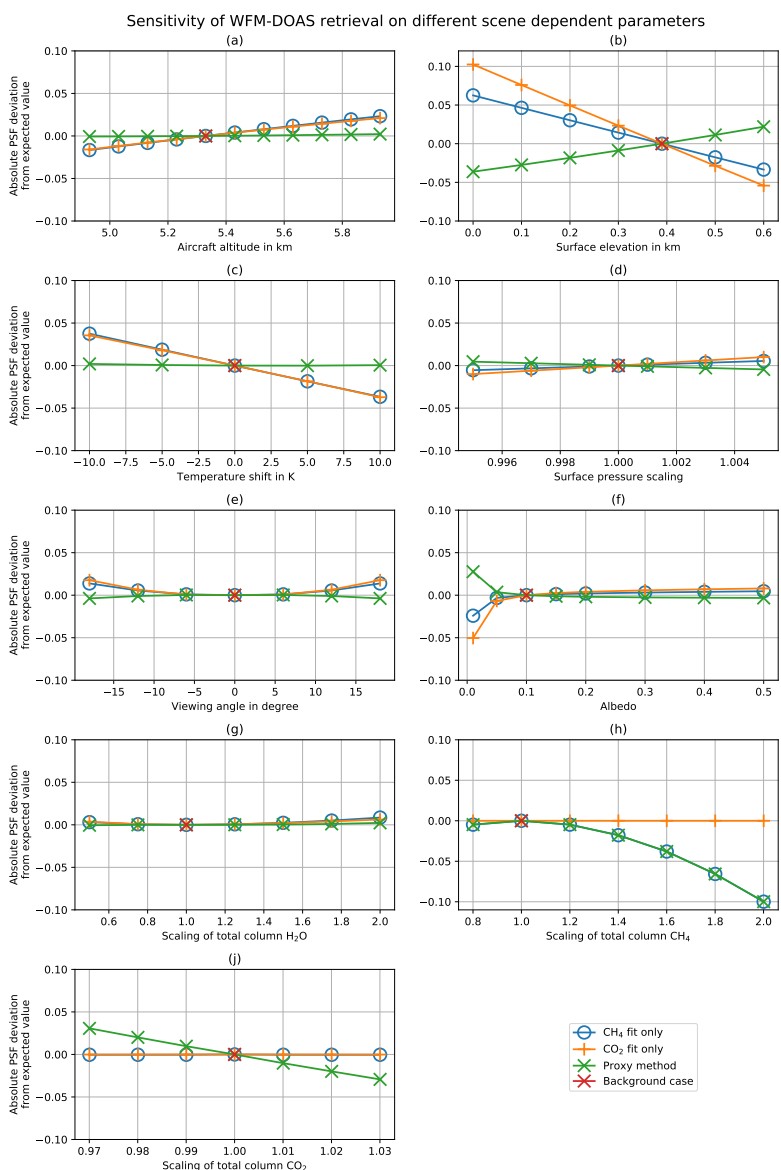

**Figure 4.** Sensitivity analysis of WFM-DOAS to the examined input parameters of the SCIATRAN radiative transfer calculation. The absolute deviation of the retrieved from the expected $PSF_{\mathrm{CH_4}}$ (blue), $PSF_{\mathrm{CO_2}}$ (orange) and $PSF_{\mathrm{CH_4,proxy}}$ (green) is plotted for each parameter. The largest deviations are present in surface elevation and scaling of the total column of $\mathrm{CH_4}$. With the proxy method, the deviations are reduced for all parameters except for a scaling in the total columns. For enhancements of $\mathrm{CO_2}$, the proxy is worse than the single $\mathrm{CH_4}$ retrieval due to division by the amount of $\mathrm{CO_2}$ (see also Sect. 3.3).





**Table 3.** Uncertainty estimate of $PSF_{CH_4}$, $PSF_{CO_2}$ and $PSF_{CH_4,proxy}$ due to the assumption of a constant albedo over different surfaces. As long as there is a constant surface type, a large value does not principally hinder source detection. However, especially paving or man made structures vary spatially with non man made structures, so false positive hits due to surface reflectance are possible.

| Surface type | Uncertainty on $PSF_{CH_4}$ | Uncertainty on $PSF_{CH_2}$ | Uncertainty on $PSF_{CH_4,proxy}$ |
|---|---|---|---|
| Rangeland vegetation | 0.38% | 0.34% | 0.04% |
| Barbed goat grass | −7.16% | −0.06% | −7.11% |
| Soil (Entisol) | 0.26% | 0.79% | −0.53% |
| Grey sandstone | 0.25% | 0.62% | −0.37% |
| Weathered aluminium | 1.30% | 1.05% | 0.25% |
| Weathered steel | −0.94% | 0.18% | −1.11% |
| Paving asphalt | 2.23% | 0.33% | 1.90% |
| Paving concrete | 11.89% | 0.77% | 11.04% |

**Table 4.** Uncertainty estimate resulting from the assumed constant atmospheric and geometric background parameters. For each parameter, the maximum deviations for $PSF_{CH_4}$ and $PSF_{CO_2}$, as well as for $PSF_{CH_4,proxy}$, are listed. For albedo, the largest value was excluded from this table (see main text). For $CH_4$, two different cases are regarded. The case $CH_4$ ($\pm 20\%$) is valid for most of the plumes and is relevant for detection of smaller sources. The extreme case ($100\%$ increase) is only relevant near very strong sources and is excluded from the averaged systematic uncertainty. The absolutely correct retrieval of $PSF_{CO_2}$ when changing $CO_2$ is due to the relatively small range of change in $CO_2$ investigated. However, this induces relatively large uncertainties for $PSF_{CH_4,proxy}$.

| Parameter | Uncertainty on $PSF_{CH_4}$ | Uncertainty on $PSF_{CO_2}$ | Uncertainty on $PSF_{CH_4,proxy}$ |
|---|---|---|---|
| Aircraft altitude | ±2.3% | ±2.1% | ±0.2% |
| Surface elevation | ±6.3% | ±10.2% | ±3.6% |
| Temperature shift | ±3.8% | ±3.7% | ±0.2% |
| Surface pressure | ±0.5% | ±1.0% | ±0.5% |
| Viewing angle | +1.4% | +1.8% | −0.4% |
| Albedo | ±0.5% | ±0.8% | ±0.4% |
| Water vapor | +0.9% | +0.6% | +0.2% |
| $CH_4$ ($\pm 20\%$) | −1.0% | ±0.0% | −1.0% |
| $CH_4$ (extreme) | −10.0% | ±0.0% | −10.0% |
| $CO_2$ | ±0.0% | ±0.0% | ±3.1% |
| Aerosol scenario | ±0.1% | ±0.3% | ±0.2% |
| Systematic uncertainty | ±8.0% | ±11.3% | ±5.4% |

surface reflectance will remain. Paving concrete would be especially likely to cause a false positive, since it induced a large positive bias. However, this would be highly correlated to structures visible in the RGB images of the scene. Barbed goat grass,



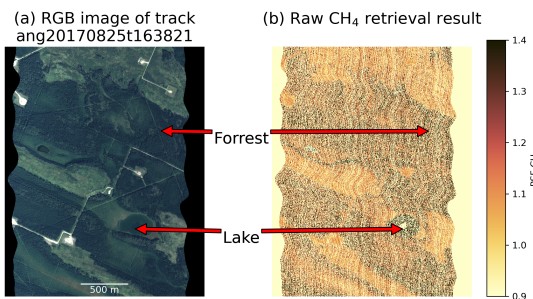

**Figure 5.** Raw retrieval results of CH$_4$ (pure $PSF_{CH_4}$, not filtered and destriped, no proxy method applied) over a scene with large forest areas and a lake as an example of a dark scene. The noise in the profile scaling factors results from the retrieval not being able to distinguish between absorption features of CH$_4$ and surface reflectance due to the low signal and therefore low signal to noise ratio. This effect is especially strong over water, but also significant over the forest areas.

on the other hand, led to a large underestimation of the total column. However, this surface type normally occurs over large patches of land, so that local enhancements on top of this bias should be detectable in most cases.

## 3.4 Filtering of poor fits

With the estimate of the influence of the background assumptions in place, we performed radiative transfer calculations for the different flight tracks and applied the retrieval to the whole data set. Examining the data it was obvious, that the retrieval sometimes failed to retrieve meaningful results. Especially over surfaces with low spectral reflectance, and therefore low signal on the detector, it produced mostly noise with profile scaling factors ranging from below 0 to largely over 2 and dramatic changes between neighboring ground scenes (see Fig. 5). This effect, due to the low SNR over dark surfaces, indicated the need to filter out low-radiance ground scenes. For IMAP-DOAS, Ayasse et al. (2018) concluded in a simulation study, that at sensor radiances below $0.1\,\mu\mathrm{W\,cm^{-2}\,nm^{-1}\,sr^{-1}}$ in the background signal led to significantly more inaccurate estimate of the methane column. In this study, we analyzed measured radiance spectra to estimate the radiance below which the retrieval results were not trustworthy.

The threshold was determined by following procedure: For each ground scene the difference between the measured and the fitted spectrum was calculated for each spectral pixel after the retrieval. These values were added in quadrature to get the root mean square difference between fit and measurement (RMS). This RMS value was then plotted over the radiance at $2140.0\,\mathrm{nm}$ in box plots with $0.05\,\mu\mathrm{W\,cm^{-2}\,nm^{-1}\,sr^{-1}}$ bins on the horizontal axis (Fig. 6) for the whole data set. For low radiances this difference increased drastically, implying a strongly reduced fit quality. As a compromise between coverage and quality, we introduced a threshold of $0.1\,\mu\mathrm{W\,cm^{-2}\,nm^{-1}\,sr^{-1}}$. The filter rejected all retrieval results where the radiances at $2140.0\,\mathrm{nm}$ were below this value. We also rejected measurements with an RMS over $2\,\%$ to remove the worst outliers. Interestingly, for very bright surfaces the spread of the upper whisker, denoting the 75 to 95 percentile, is increased. This could have been from surfaces such as paving materials or other anthropogenic structures, for which the reflected spectrum already had interfering





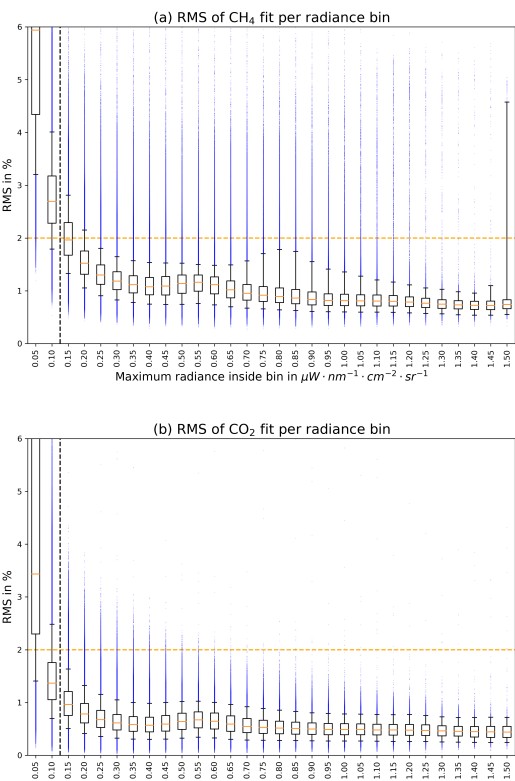

**Figure 6.** Difference between fit and measurement (RMS) over radiance at $2140.0\,\mathrm{nm}$. The box indicates the first to third quartile range, the whiskers denote the 5 to 95-percentile, and the small orange line inside the box the median RMS value of the according radiance bin. The small blue dots denote outliers outside the 95 percentile. For low radiances, the fitting quality decreases significantly, as shown in the high values and wide spread of the RMS. Therefore, all measurements over surfaces with radiances below $0.1\,\mu\mathrm{W\,cm^{-2}\,nm^{-1}\,sr^{-1}}$ are filtered out (dashed black vertical line), as the results are unreliable. Results with RMS higher than $2\,\%$ are filtered out as additional quality flag (dashed orange line). The increase in RMS spread especially at $1.5\,\mu\mathrm{W\,cm^{-2}\,nm^{-1}\,sr^{-1}}$ might be due to high reflecting structures, which have broad band reflection features similar to absorption features of $CH_4$ (see Ayasse et al., 2018, and Sect. 3.3).

features similar to the absorption of $CH_4$ at the AVIRIS-NG spectral resolution (see also Sect. 3.3 and Table 3). This results agree with the findings of Ayasse et al. (2018).



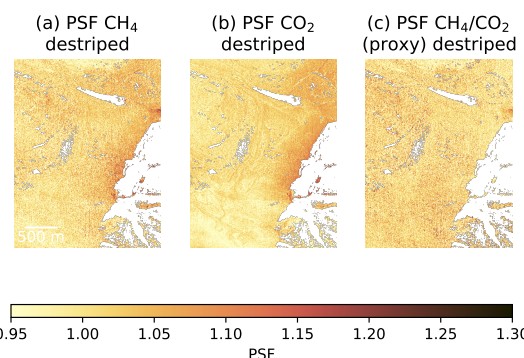

**Figure 7.** Example of the effect/necessity of the proxy method from flight line ang20170823t180156. In (a), the destriped $PSF_{CH_4}$ is shown. Simply analyzing this image, one could assume a diffuse enhancement due to perhaps coal mining activities, as this is part of an open cast coal mine. In figure (b) the destriped $PSF_{CO_2}$ is shown. There, a similar enhancement is visible. In (c) the destriped proxy results are shown, where this diffuse enhancement vanishes.

## 4 Detection and inversion of plumes

### 4.1 Detection of plumes

For the detection of plumes, we filtered the retrieved $CH_4$ values (Sect. 3.4), removed striping (Sect. 3.2 and Sect. A in the appendix) and calculated the final $CH_{4,enh}$ according to Eq. 2. The proxy method proved necessary; otherwise diffuse $CH_4$
enhancements would be mistaken for true enhancements due to emissions of $CH_4$. This can be seen in Fig. 7, where diffuse enhancements in the pure $CH_4$ results vanished completely after applying the proxy method.

The final $CH_{4,enh}$ were plotted as images and were manually inspected for methane plumes. For the reduced data set used in this study (13 tracks), this approach detected several plumes in 10 out of the 13 tracks. However, many plumes were faint or located near infrastructure, making unambiguous detection difficult and/or preventing the application of the cross sectional
flux method (White et al., 1976, and Sect. 4.2). Therefore, we show plumes which are well suited for the cross sectional flux method in Fig. 8 and Fig. 9. The additional $CH_4$ plumes and enhancements not inverted in the following sections can be found in the appendix in Sect. B. Those comprise, among others, emissions most likely resulting from open cast coal mining (Fig. B1) or a well pad located in a forest (Fig. B2).

In Fig. 8, two overpasses of the same source on two different days are shown. On the first day (Fig. 8 (b)), the plume structure
was recognizable over a relatively long distance, while on the second day (Fig. 8 (d)), the plume was only faintly visible in the vicinity of the source. This was most likely due to the wind speed, which was significantly higher on the second day ($4.4\,\mathrm{m\,s^{-1}}$) compared to the first day ($2.6\,\mathrm{m\,s^{-1}}$). A wind speed higher by a factor of 1.6 means a decrease in the column enhancements by a factor of 1.6, which reduced the visibility of the plume in the retrieval results.





In Fig. 8 (b), a new interesting feature was observable at the source. There, we observed a double plume structure that was especially prominent during the first overpass. Comparison with the RGB image revealed, that one part seemed to originate from the vent, while the other part seemed to originate from the top of the shadow of the vent. The vent released the plume several meters above the surface. Because the plume was very narrow near the source, the sunlight only passed the plume

either before or after hitting the ground. As those two light paths were attributed to different ground scenes, the absorption and therefore the apparent $CH_4$ enhancements were visible at two locations leading to the double plume structure. Further down the plume, atmospheric mixing took place and the plume widened. Then, the sunlight passed through the plume both before and after hitting the ground, and the double plume structure vanished. A simple geometric consideration of the distance between the two plume structures, the solar and instrument zenith angle and the vent height estimated from the shadow of the structure

supported this hypothesis.

In Fig. 9, two plumes originating from well pads are shown. Both extended linearly from their source and were visible over approximately $100\,\mathrm{m}$. While the first plume (Fig. 9, b) originated from a cold vent (similar to Fig. 8), the emitting structure for the plume in Fig. 9 (d) could not be identified from the RGB images. It seemed, however, that the source was located near the surface. This would also explain the large deviation of the plume direction from the wind direction aquired from the ERA-5

model data, since the nearby forests could have significantly altered the wind direction near the surface.

## 4.2 Cross sectional flux method

To estimate the fluxes for the selected sources, we applied the cross sectional flux method (White et al., 1976). As this method is computationally and conceptually simple, it could well be used for initial estimations of source strengths. It had successfully been utilized for estimating of emissions detected in remote sensing measurements (for example in Krings et al., 2011;

Frankenberg et al., 2016; Krautwurst et al., 2017; Varon et al., 2019). In this method, one calculates the flux $F$ in kilogram per hour through a transect orthogonal to the wind direction with length segments $\mathrm{d}x_i$ in meter based on the total column enhancements $CH_{4,\mathrm{enh},i}$ in molecules per square centimeter at position $i$ along the transect:

$$F = f \cdot \sum_i CH_{4,\mathrm{enh},i} \cdot u \cdot \mathrm{d}x_i. \tag{3}$$

The wind speed $u$ in meter per second is assumed to be constant in time and space for the time of overflight. For the detected

AVIRIS-NG plumes, this assumption was valid, as these plumes had been sampled within seconds. The wind speed was extracted from the ECMWF ERA5 data (see Sect. 2.2), while the wind direction was estimated from the observed plume directions. The factor $f = 9.589e-23\,\mathrm{s\,kg\,h^{-1}\,molec^{-1}}$ converted the flux to kilogram per hour.

We defined the local background for $CH_{4,\mathrm{enh}}$ for each cross section as the region outside of the plume on each side. Then, we calculated the $PSF_{\mathrm{proxy,bg}}$ for the normalization to the local background using a linear fit between both local background

regions. This background fit reduced slight gradients present in the background concentration to accurately estimate the column enhancements originating from the source.

As is observable in Fig. 8, there were gaps and accumulations along the plume. These were caused by eddies and short gusts which disrupted the plume structure. To account for that atmospheric variability, we defined multiple cross sections along the



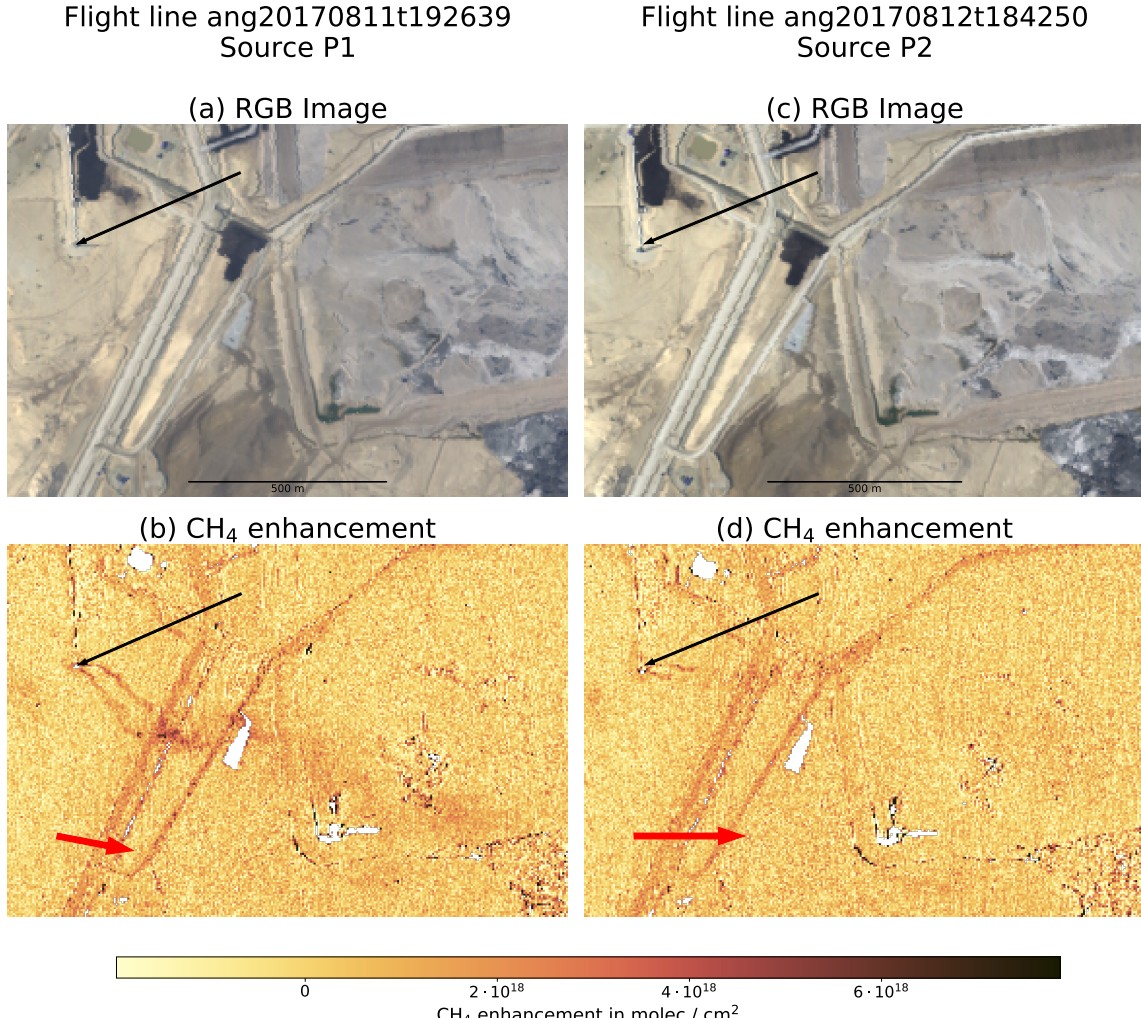

**Figure 8.** Plume resulting from a cold vent. The black arrow denotes the source position, while the red arrow indicates the wind direction and wind speed according to ERA-5 data for comparison to the plume direction. In (b) the plume is much fainter, most likely due to the higher wind speed. Especially in (a) a double plume structure is visible. This structure likely results from the light passing through the plume only once before reflection on the ground for one part of the double structure, and only once after reflection for the second. As vertical and horizontal mixing takes place the further the emissions travel, the light passes through the plume twice, and the double plume structure vanishes. The roads are prominently visible in the retrieval results. It might be that this road is made of concrete, or otherwise contains limestone. The roads were excluded from the flux analysis due to the high bias induced by it.

plume, each one pixel apart. We then calculated the flux for each of the cross sections. The final flux estimate was the mean value of the single fluxes through all cross sections.




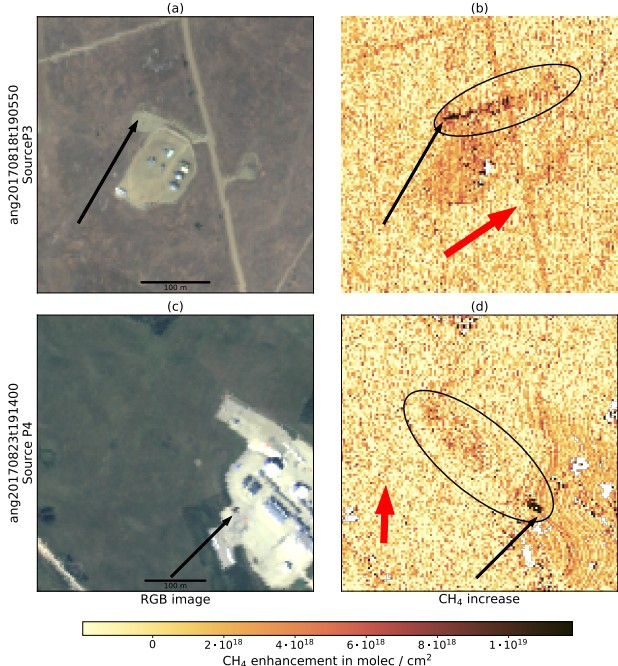

**Figure 9.** Two additional plumes detected in the retrieval results, emanating from well pads. On the left ((a) and (c)), the RGB images obtained from radiances of AVIRIS-NG are shown. On the right ((b) and (d)), the according retrieval results are presented. The ellipse highlights the plumes. Additionally, the black arrows point to the source of the plumes, while the red arrow indicates the wind direction and wind speed according to ERA-5 data for comparison to the plume direction. The sources are located near grass land. One can see the plume shooting straight away in both cases. For (d) however, the ERA-5 wind direction does not fit well to the plume. This might result from the plume originating from a source very near to the ground, where the influence of forests or small hills largely influences the wind direction. Additionally, near the ground the instantaneous wind direction resulting from reanalysis data is uncertain and could explain the difference.

### 4.3 Flux and uncertainty estimation based on cross sectional flux method

For the plumes shown in Fig. 8 (in the following P1 and P2 for (b) and (d) respectively) and Fig. 9 (P3 and P4 for (b) and (d) respectively), the methane flux was calculated using the cross sectional flux method. We selected plumes P1 and P2 for two reasons: P1 was visible for approximately $200\,\mathrm{m}$ before crossing the road, making it possible to define multiple cross sections

5     through the plume and thus leading to a strong reduction in the uncertainties. This was the only source observed twice in an emitting state by AVIRIS-NG, which allowed comparison of the flux estimates for two overflight times. It originated from a vent in a bitumen extraction site. P3 and P4 shown in Fig. 9 (b) and (d) also showed a well shaped straight plume, which was favorable for the cross sectional flux method. In Fig. 8 (b) and Fig. 9 (b), a clear bias due to the underlying road surface was visible. In both cases, cross tracks that overlapped with this bias were excluded from the flux estimation.

10     Application of the cross sectional flux method on all plumes yielded the following emission estimates: For P1, a mean flux of $(141 \pm 87)\,\mathrm{kg\,h^{-1}}$ was calculated, while for P2, the mean flux was $(89 \pm 46)\,\mathrm{kg\,h^{-1}}$. For P3 and P4, the mean fluxes were





$(111 \pm 76)\,\mathrm{kg\,h^{-1}}$ and $(70 \pm 79)\,\mathrm{kg\,h^{-1}}$, respectively. We calculated the wind speeds for the different plumes to be $2.7\,\mathrm{m\,s^{-1}}$ for P1, $4.4\,\mathrm{m\,s^{-1}}$ for P2, $2.8\,\mathrm{m\,s^{-1}}$ for P3 and $1.5\,\mathrm{m\,s^{-1}}$ for P4.

We estimated the uncertainty of the flux via Gaussian error propagation from the uncertainties of the single components of Eq. 3 and Eq. 2, with additionally accounting for the atmospheric variability. The contribution and derivation of the main error
sources for P1 is shown below. We used the same procedure to estimate the uncertainty for the other sources.

We estimated the single pixel precision for each plume region by analyzing the local background of the plume. We used the standard deviation of the local background $PSF_{\mathrm{CH_4,proxy}}$ as measure of the retrieval noise and included the uncertainty due to small variations of $CO_2$, which might had been coemitted from the flare or collocated to the plume but which were below the noise of the pure $PSF_{\mathrm{CO_2}}$. This led to a single pixel precision of $\sim 3\,\%$ of the $CH_4$ background column, which translated to an
uncertainty on the final flux of $\pm\,26\,\mathrm{kg\,h^{-1}}$.

The uncertainty due to atmospheric stability was calculated as the confidence interval of the fluxes through all cross sections. This amounted for an uncertainty on the final flux of $\pm\,24\,\mathrm{kg\,h^{-1}}$.

For the the wind, we assumed an uncertainty of $\pm\,1.5\,\mathrm{m\,s^{-1}}$, according to the standard deviation of the hourly wind measurements of ERA5 compared to inland measurement stations (Minola et al., 2020). For most cases the wind direction matched
quite well, so we assumed that the wind speed was also reasonably well captured by the model. As the wind speed directly influenced the flux, and the uncertainty could not be reduced by simply taking more cross sections into account, this directly propagated to an uncertainty of the final flux of $\pm\,79\,\mathrm{kg\,h^{-1}}$.

The estimate of the background column was dependent on a scaled climatology. We assumed a $\pm\,5\,\%$ deviation as an upper limit of the uncertainty for the total column of $CH_4$ around the plume compared to this climatology. This lead to an uncertainty
of the final flux of $\pm\,7\,\mathrm{kg\,h^{-1}}$, which was very small compared to the other uncertainties.

Assuming that those sources of uncertainty were uncorrelated, we combined them in quadrature. This resulted in an uncertainty for P1 of $\pm\,87\,\mathrm{kg\,h^{-1}}$. An overview over the contribution of the single error sources for all sources is given in Table 5. We emphasize that for all of these cases, the fluxes were calculated from snapshots and were only valid for the time of overflight.

Our estimates for P1 and P2 overlapped within their respective uncertainties, suggesting that the source may have been
approximately constant over the two days. This could be explained by uncertainties in our assumptions of the wind speed, and the stronger dilution of the plume on the second day (Fig. 8). Additionally, the ratio between the mean fluxes (P1/P2) was nearly the same as the inverse ratio of the wind speeds ($4.4\,\mathrm{m\,s^{-1}}/2.7\,\mathrm{m\,s^{-1}}$), which further supported the theory, that dilution due to the higher wind speed played an important role for the lower flux on the second day.

The plume P4 was observed under unfavorable wind conditions with low winds of $1.5\,\mathrm{m\,s^{-1}}$. This led to an uncertainty of
more than $100\,\%$. For such low winds, however, the cross sectional flux method is not well suited, as there horizontal transport due to diffusion is completely neglected. This however plays an increasing important role the lower the wind speed is (Sharan et al., 1996).





**Table 5.** Inversion results and uncertainty estimate based on the cross sectional flux method for P1 and P2 (two overpasses of the same source on two consecutive days) and P3 and P4. The single pixel precision was calculated from the standard deviation of a background region for each plume. The atmospheric variability resulted from the confidence interval of the multiple cross sectional fluxes. The wind uncertainty and total column uncertainty was assumed to be $\pm 1.5\,\mathrm{m\,s^{-1}}$ and $5\,\%$, respectively (see also text for explanation). For all inversions, the wind speed uncertainty is a very large uncertainty. While for P2 additionally the atmospheric variability induces similar large errors, for P3 and P4 this is much less the case. However, there the different surfaces lead to a much larger single pixel uncertainty.

|  | P1 | P2 | P3 | P4 |
|---|---|---|---|---|
| Single pixel precision | $\pm 26\,\mathrm{kg\,h^{-1}}$ | $\pm 21\,\mathrm{kg\,h^{-1}}$ | $\pm 40\,\mathrm{kg\,h^{-1}}$ | $\pm 27\,\mathrm{kg\,h^{-1}}$ |
| Atmospheric variability | $\pm 24\,\mathrm{kg\,h^{-1}}$ | $\pm 24\,\mathrm{kg\,h^{-1}}$ | $\pm 20\,\mathrm{kg\,h^{-1}}$ | $\pm 12\,\mathrm{kg\,h^{-1}}$ |
| Wind uncertainty | $\pm 79\,\mathrm{kg\,h^{-1}}$ | $\pm 26\,\mathrm{kg\,h^{-1}}$ | $\pm 59\,\mathrm{kg\,h^{-1}}$ | $\pm 72\,\mathrm{kg\,h^{-1}}$ |
| Total column uncertainty | $\pm 7\,\mathrm{kg\,h^{-1}}$ | $\pm 4\,\mathrm{kg\,h^{-1}}$ | $\pm 6\,\mathrm{kg\,h^{-1}}$ | $\pm 4\,\mathrm{kg\,h^{-1}}$ |
| Total flux | $(141 \pm 87)\,\mathrm{kg\,h^{-1}}$ | $(89 \pm 46)\,\mathrm{kg\,h^{-1}}$ | $(111 \pm 76)\,\mathrm{kg\,h^{-1}}$ | $(70 \pm 79)\,\mathrm{kg\,h^{-1}}$ |

## 5 Discussion

The WFM-DOAS retrieval provided an efficient and accurate way to handle AVIRIS-NG data quantitatively. In contrast to the IMAP-DOAS retrieval (Frankenberg et al., 2005; Thorpe et al., 2014), the WFM-DOAS retrieval is a non-iterative retrieval with precalculated radiative transfer calculations. This reduced the computational time needed for the retrieval, while still
delivering reliable local total column enhancements. On the other hand, the modeled background spectrum is adapted to the physical properties of the scene by scaling the trace gas columns and adapting the geometric parameters necessary to model the average light path over the scene, in contrast to the more statistical approach used in the matched filter (Thompson et al., 2015).

     The WFM-DOAS retrieval of $CH_4$ in the two SWIR fit windows of AVIRIS-NG data produced much larger noise in the
results for the weak window than for the strong window. The significantly less noisy WFM-DOAS results around $2300\,\mathrm{nm}$ most likely originated from the higher number of spectral data points for the fit, and the stronger absorption features. Even though there was approximately half the amount of light reaching the detector at these wavelengths, which reduced the SNR on the detector, the SNR was still high enough for a good retrieval.

     Problems however arose over very dark surfaces or surfaces with reflection properties similar to absorption features of $CH_4$
at the spectral resolution of AVIRIS-NG. This led to residual structures in the retrieved $CH_4$ maps. Especially paved roads or other anthropogenic structures were observable. Even though the $CH_4/CO_2$ proxy method reduced false positives in many cases, there was still a remaining dependency of the $CH_4$ results from the surface spectral reflectance for some surfaces like concrete or barbed goat grass. Additionally, the noise on the retrieval results varied over different surfaces, which is reflected in the uncertainties for the flux inversions for the 4 plumes.



Those effects could be mitigated by deploying and utilizing an imaging spectrometer specifically designed for the task of monitoring $CH_4$ and $CO_2$ concentrations, such as the proposed Airborne Methane Plume Spectrometer (AMPS Thorpe et al., 2016b), or the MAMAP 2D system currently being developed and built at the University of Bremen, Germany. Due to the higher spectral resolution, those instruments will have a higher sensitivity to smaller enhancements and should be less

influenced by the surface reflectance properties, as is already the case for the MAMAP instrument (see e.g. Krings et al., 2011).

A large uncertainty in the flux inversion of emission sources, which can not be solved by advancing the imaging remote sensing instrument's characteristics, arises from the wind speed estimation. Additional measures have to be taken to reduce this uncertainty. For example, in situ wind measurements in the boundary layer could additionally be made (see e.g. Krautwurst et al., 2017; Krings et al., 2018). This approach is especially useful when airborne in situ measurements are included in the

campaign design. Another possibility includes the deployment of wind lidars (Wildmann et al., 2020), similar to the approach taken in the $CO_2$ and Methane mission (CoMet). However, for large surveys or transects, this is not feasible anymore, especially when the source locations of the plumes are not known prior to the flights. There, either advancing to local wind models with much higher spatial resolution such as the GRAL model (Berchet et al., 2017) or the MECO(n) model (Kerkweg and Jöckel, 2012) could lead to a significant uncertainty reduction. Also, methods such as the integrated mass estimation (IME,

Jongaramrungruang et al., 2019), which use empirically derived correlations between surface wind, flux rates, plume shape and mass enhancement in the plume to estimate the wind speed and the flux, could help estimating the emissions independent from local wind models or wind measurements. Additionally, more sensitive remote sensing instruments could observe the plume over longer distances, where the plume is likely better mixed in the boundary layer and the horizontal extend of the plume is less influenced by turbulence and gusts, so that the modeled wind speed most likely better matches the wind speed inside the

plume.

## 6 Summary and Conclusions

We successfully adapted and applied the WFM-DOAS retrieval to AVIRIS-NG data and estimated the uncertainties of this method. In the data set, we were able to detect several point sources. An estimation of the methane emissions of a vent revealed emissions of $(141 \pm 87)\,kg\,h^{-1}$ and $(89 \pm 46)\,kg\,h^{-1}$ on two consecutive days, while two other sources related to gas extraction

emitted $(111 \pm 76)\,kg\,h^{-1}$ and $(70 \pm 79)\,kg\,h^{-1}$, respectively. These source strengths are quite common as indicated by the log normal distribution of sources in the Four Corners region (Frankenberg et al., 2016). A large source of uncertainty for the flux inversion was the wind speed estimate, as no collocated wind speed measurements near the surface were collected. Also the atmospheric variability played an important role for shorter (i.e. smaller) plumes. This influence is reduced for longer plumes, as more tracks are available for the flux estimation. The noise on the retrieval results varied with different surfaces,

which contributed notably to the uncertainty of the flux estimate. For the high wind situation, parts of the plume might have been additionally missed due to the higher dilution.

The dependency of the resulting total column $CH_4$ retrieval results from the parameter values assumed in the radiative transfer calculation have been examined. For most parameters, the induced bias was reduced to well below $1\,\%$ when using $CO_2$



as a proxy for light path correction. Large perturbations in elevation resulted in a residual bias, however, the elevation varied mostly smooth (for example over hills) or only by smaller amounts over buildings or vegetation changes such as from grassland to forests. In addition, very strong $CH_4$ enhancements led to a systematic underestimation of $CH_4$, which in consequence could lead to an underestimation of very strong emitters. However, such strong enhancements only occur near to strong sources (not

present in this study) and therefore inversion estimates could be performed further down the plume, where the enhancements are lower and therefore the bias is negligible. Deviations of $CO_2$ from the background on the other hand were retrieved correctly for typical variations of the total column. However, due to the use of $CO_2$ as proxy for light path correction, these deviations led to a bias in the proxy. Consequently, large amounts of $CO_2$ coemitted to $CH_4$ may mask a weak $CH_4$ plume. Additionally, the influence of some surface reflectance spectra on the retrieved $PSF_{CH_4,proxy}$ was examined. While for some surface types the

bias on the retrieved $PSF_{CH_4,proxy}$ could be reduced to well below $1\%$, some surfaces introduced larger biases, reaching up to $11\%$ for paving concrete.

As dark surfaces mostly produced noise in the retrieval results, ground scenes with at sensor radiances below $0.1\,\mu W\,cm^{-2}\,nm^{-1}\,sr^{-1}$ were excluded from the analysis. In the future, this radiance filter described in Sect. 3.4 could be applied to the data before the application of the WFM-DOAS retrieval. This reduces the amount of data which has to be retrieved, without rejecting possible

good retrieval results. Additional retrieval improvements could be achieved by fitting $CH_4$ and $CO_2$ in both the weak and the strong window simultaneously.

While this and previous studies have demonstrated detection and quantification of methane emission sources with AVIRIS-NG, the residual structures due to the relatively coarse spectral resolution make unambiguous detection and especially quantification of small sources difficult. To mitigate this problem, spectrometers dedicated to the detection and quantification of $CH_4$

are currently developed, such the proposed AMPS system (Thorpe et al., 2016b) or the MAMAP-2D system, which is being assembled.

*Data availability.* The AVIRIS-NG data set of the ABOVE campaign is available at https://daac.ornl.gov/cgi-bin/dsviewer.pl?ds_id=1569 (Miller et al., 2019a).

## Appendix A: Striping Effect

In a push-broom imager such as AVIRIS-NG each detector column acts as one separate line detector looking at a different ground scene. Even with very good calibration and characterization (described in Chapman et al., 2019, for AVIRIS-NG), there will still be small differences in the spectra recorded by two different lines. In the retrieval results, this leads to a line (or column) dependent difference in the retrieval results called "striping" (see for example Fig. 5). To correct this effect, we normalize the $PSF_{CH_4}$, $PSF_{CO_2}$ and $PSF_{CH_4,proxy}$ for each pixel by the median PSF of its corresponding detector column. We

select the median for resilience against outliers, which could otherwise have a large impact on the correction


## Appendix B: Other detected CH$_4$ plumes

In this section, additional plumes are shown, which have been detected in the data, but were either too inconsistent or the conditions did not allow for the application of the cross sectional flux method.

A very interesting observation are the plumes and CH$_4$ accumulations due to open cast coal mining, as can be seen in Fig. B1.
5  During coal mining, the enclosed methane is released to the environment. For underground mining, the methane concentration in the air in the mining shafts are kept well below the explosion limit by ventilating the shafts with fresh air (Özgen Karacan, 2008). The ventilation shafts then emit the total emissions from the whole mine. In an open coal mine the emissions may come out diffuse, however, according to Fig. B1, it seems as significant amounts of methane may be emitted from the brim, perhaps during cutting of the coal.

10  Additional plumes are shown in Fig. B2 and Fig. B3. Those plumes originate from oil and gas infrastructure such as a well pad in the forest and vents. Especially the vents in B3 are only very faintly visible.

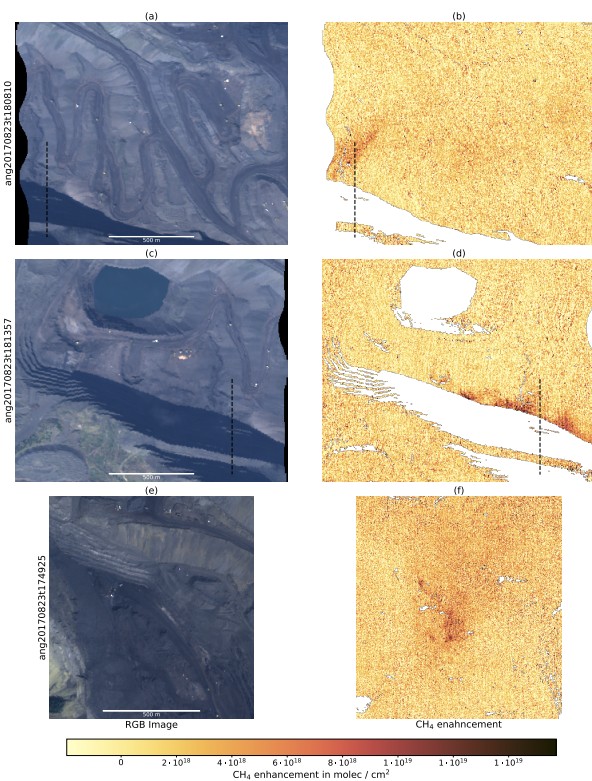

**Figure B1.** Methane enhancements resulting from bituminous coal extraction. On the left (a, c, e) the RGB images and on the right (b, d, f) the according total column enhancements of $CH_4$ are shown. The upper two measurements where taken within $\sim 10$ minutes, and the flight lines lay very close to each other, so that the brim is the same. The dashed line is plotted on the same location in both images. While in (b) a plume emanating from the brim is visible, in (d) strong accumulations near the brim are visible. In (f) the plume is more diffuse and the highest enhancements are located near a brim. However, it is not as clear as in (b) and (d), if it really originates from the brim.



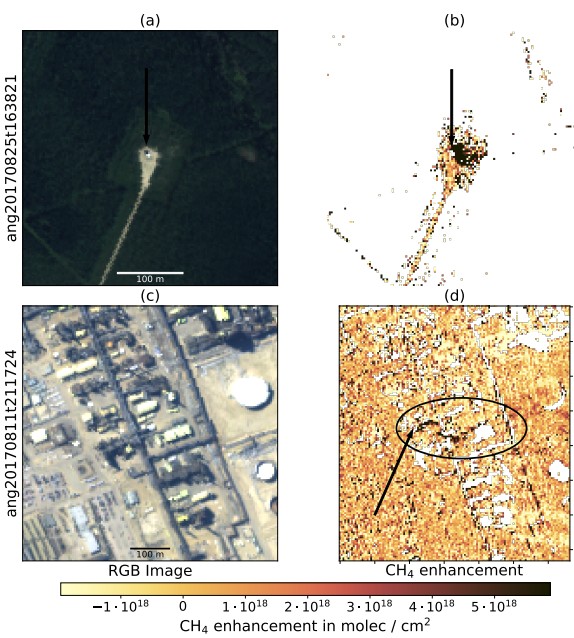

**Figure B2.** Similar to Fig. B1, but for methane enhancements from oil/gas infrastructure. In (a) and (b), the emissions originate from a well pad in a forest. Due to the low radiance over the trees, only the well pad itself passes the quality filters. The enhancements seem accumulated and no clear wind direction is visible. In (c) and (d), a facility located at a bitumen extraction site is shown and meanders through the facility.





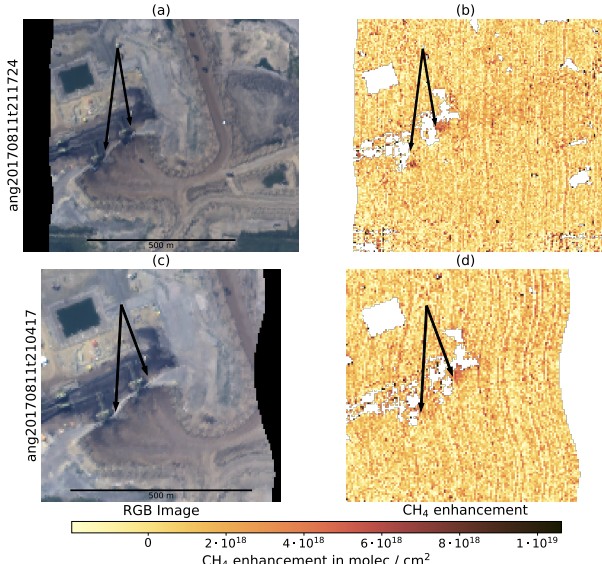

**Figure B3.** Similar to Fig. B1, but for two overpasses over cold vents at a bitumen extraction site. The plumes are only faint, especially for the second overpass (d). In the results a small striping effect is still visible. The destriping reduces the effect, but is not able to totally eliminate it.

*Author contributions.* JB contributed to the study design, adapted the retrieval, analyzed the data and wrote the manuscript, KG and HB initialized the study, KG and SK contributed to the retrieval adaption, study design and paper draft, AKT, DRT and CF helped in data set selection and with the handling of the data, RD and CM designed the flight plans, organized and led the flights, HB and JPB supervised the work. All authors contributed to the final manuscript.

5  *Competing interests.* The authors declare that they have no conflict of interest.

*Acknowledgements.* This work was funded in parts by the BMBF project AIRSPACE MAMAP2D (FKZ01LK1701B). We would also like to acknowledge the contributions of the AVIRIS flight and instrument teams. The AVIRIS-NG data were collected as part of the Arctic Boreal Vulnerability Experiment (ABoVE), a NASA Terrestrial Ecology program, and other methane studies supported by NASA's Earth Science Division. A portion of this work was performed at the Jet Propulsion Laboratory, California Institute of Technology, under contract with
10  NASA.





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
