# Peer review of "Detection and Quantification of $CH_4$ Plumes using the WFM-DOAS retrieval on AVIRIS-NG hyperspectral data"

_Atmospheric Measurement Techniques, 2020_

## Referee Comment (RC1) · Anonymous Referee #2 · 23 Sep 2020

**General comments**

Borchardt et al. present an application of the weighting function modified differential optical absorption spectroscopy (WFM-DOAS) trace gas retrieval algorithm to AVIRIS-NG aircraft remote sensing observations. They evaluate the ability of the algorithm to retrieve methane and carbon dioxide column concentrations from the aircraft data, and demonstrate its effectiveness for detecting and quantifying individual methane point sources. The authors present comprehensive error analyses for both the column retrieval and their emission rate estimates, and discuss ways in which the uncertainties could be reduced in the future. The paper is a good fit for *AMT*. I would recommend

publication subject to minor revisions as outlined below.

**Specific comments**

- Please define the WFM-DOAS acronym early on in the text.

- The abstract is very long. Much of the content there seems better suited for the introduction. I would recommend shortening the abstract to one or two paragraphs so the reader can quickly get a sense of the paper's main findings/results.

- P4, L11: Hamlin et al. (2011) cite 10 nm spectral sampling for AVIRIS and 5 nm spectral resolution for the next-generation instrument. I can't find a statement of 5 nm spectral sampling in their paper. If I'm not mistaken, can you add a reference for the 5 nm spectral sampling number?

- P5, L2: What kind of interpolation is used?

- P5, L3: Any motivation for using the 50-m average? Is it just a convenient choice or is there an argument for this being the most relevant wind speed?

- P5, L4: I couldn't find a description in the text of how the wind direction was assessed from the plume shape. Can you elaborate?

- Table 1: Can you explain why the PSF-CH4-proxy uncertainty is (slightly) higher than the PSF-CH4 uncertainty? Error stacking from the uncorrelated errors in PSF-CH4 and PSF-CO2?

- I'm familiar with the CO2 proxy method being applied in the 1650 nm window where there is also strong CO2 absorption, but not in the 2300 nm window. Can you comment on the effectiveness of the method at 2300 nm compared to 1650 nm?

[Figure]

- Can you explain more clearly the pros/cons of WFM-DOAS compared to IMAP-DOAS and Matched Filter? How much faster is WFM than IMAP? When/why is it better than Matched Filter? You say in the abstract that WFM "fills a gap" between the other two retrieval algorithms, and mention some of the differences between them in section 1 and section 5, but I didn't get a clear picture of the motivation behind WFM, except that it's computationally cheaper than IMAP. What are the trade-offs? A concise explanation in the introduction of when/why we should use WFM instead of IMAP or MF would be very helpful.

- P18, L1-10: The double plume structure is intriguing. Could you briefly elaborate on the height of the source, solar zenith angle, and distance between the two plume images in this case? Also, how did you determine wind direction and do the cross-sectional method for this plume?

- P18, L28: How do you define inside/outside the plume? I think previous AVIRIS studies used a threshold, did you do that too?

- P21, L1: Is the wind speed calculation referred to here the 50-m average from ERA5, or something else?

- Section 4.3: Are these 1-sigma or 2-sigma errors? Or confidence intervals?

- P21, L26: I don't understand why the higher wind would cause you to underestimate the emission rate. Shouldn't the cross-sectional method be able to deal with this? Each plume transect gives you an emission estimate, and the higher wind speed will result in higher estimates. I can see how the precision would be lower for a weaker plume, since you might have fewer transects to analyze. But a negative bias would imply that you are missing part of each transect. In that case, couldn't you extend the transects further from the plume axis? This goes back to my question about defining inside/outside the plume...

**Technical corrections**

- P10, L33: Sentence unclear, please rephrase.

- Figure 5: Typo in "forest."

- Figure 8: Panel labels are wrong in the caption.

---

## Referee Comment (RC2) · Luis Guanter (Referee) · 21 Oct 2020

[Apologies for this late review!]

A number of papers in the last years have shown that measurements from the AVIRIS and AVIRIS-NG airborne spectrometers are useful to detect and quantify methane point sources. The manuscript by Borchardt et al. describes the implementation of the WFM-DOAS methane retrieval method, originally developed for satellite atmospheric spectrometers, for AVIRIS-NG data. The motivation for this implementation of the

[Figure]

WFM-DOAS method for AVIRIS-NG is to cover the gap between the potentially less accurate data-driven (particularly Matched-Filter, MF) methods and the computationally-expensive IMAP-DOAS which are typically used for this type of data.

The manuscript is well written and presented, and the topic is timely and falls perfectly within AMT's scope, so I generally recommend it for publication in AMT once the points listed below are taken into consideration.

Major comments:

1. Comparison to Matched Fiter (MF) and IMAP-DOAS methods: according to the authors, the motivation to propose a WFM-DOAS retrieval for AVIRIS-NG is to "fill a gap" between MF and IMAP-DOAS approaches. However, nothing is said about the retrieval performance of the proposed WFM-DOAS with respect to the other two methods.

I believe that the study would greatly benefit from a quantitative comparison of the three retrievals. The MF XCH4 data for the same ABOVE data set seem to be already available (p4, L20), and looking at the author list I understand that it wouldn't be difficult to also have IMAP-DOAS retrievals for at least a data subset containing one of the identified plumes.

2. Selection of the AVIRIS-NG data set used for this: the authors don't provide any explanation of why they select this particular ABOVE AVIRIS-NG data set for the study, but I would say that there could be better AVIRIS / AVIRIS-NG data sets for a study focused on the presentation of a new retrieval method. For example, the data sets from the different campaigns that JPL has run over the last years, including e.g. the one over Four Corners (Frankenberg et al., 2016) and California (Duren et al., 2019), for which the L1B data are surely available, could offer better observation conditions, properly documented plumes and fluxes, and a wider range of emission intensities and types.

So I would recommend the authors to extend the study to other AVIRIS-NG showing other acquisition conditions, and in particular larger emissions. On the other hand, if the authors were interested in the particularly challenging conditions of the Canada sites, this should be discussed in the text.

Other comments:

- Abstract: it might benefit from shortening and removal of line breaks

- p4 L25: "Copernicus", nested brackets

- Fig. 1 caption: "can not be fully resolved"

- p7, L28: plus surface reflectance shows in general less variability at 1600 nm than at 2300 nm

- Section 3.3., sensitivity analysis: it could also be relevant to include uncertainties in the spectral calibration (position and shape of the spectral response function), if this is not optimized in the retrieval

- p15, L13 "determined by the following"

- p17, L9 – the cross sectional method hasn't been mentioned until now

- p17, L17: "A wind speed higher by a factor 1.6 means..."? why is this? can you please explain this further?

- Section 4.2 and 4.3 could be merged. In general, a manuscript structure with separate Methods and Results sections would be better.

- Figure captions: perhaps just a matter of taste, but I think they could be shorter and avoid the explanations already included in the main text (e.g. Fig. 8 and 9).

- Fig. 9, Plume P4: at least from visual inspection, it is not obvious to me that that is a real plume. It could also be the effect of background reflectance variations. Would it make sense to show a map of at-sensor radiance at 2200 nm and its slope to discard

this possibility?

- Refs. At least Cusworth et al. (https://doi.org/10.5194/amt-12-5655-2019) and Foote et al. (https://doi.org/10.1109/TGRS.2020.2976888) should be added
* * *

---

## Author Comment (AC1) · 24 Dec 2020

**Author Respones to Reviewer #2 (R2)**

We would like to thank Reviewer #2 for the helpful review. In the following, we will first repeat the review comment (R2-X), and then give our response.

R2-1: Please define the WFM-DOAS acronym early on in the text.

**We added the definition of WFM-DOAS to the abstract and at its first mention in the introduction.**

R2-2: The abstract is very long. Much of the content there seems better suited for the introduction. I would recommend shortening the abstract to one or two paragraphs so the reader can quickly get a sense of the paper's main findings/results.

**We agree and shortened the abstract.**

R2-3: P4, L11: Hamlin et al. (2011) cite 10 nm spectral sampling for AVIRIS and 5 nm spectral resolution for the next-generation instrument. I can't find a statement of 5 nm spectral sampling in their paper. If I'm not mistaken, can you add a reference for the 5 nm spectral sampling number?

**We added Chapman et al. (2019) as a source for this.**

R2-4: P5, L2: What kind of interpolation is used?

**We used linear interpolation in time and space between the 4 nearest locations and the 2 nearest time steps. We added this explanation to Sect. 2.2.**

R2-5: P5, L3: Any motivation for using the 50-m average? Is it just a convenient choice or is there an argument for this being the most relevant wind speed?

**We originally used the 50 m average, as the sources P1 and P2 were estimated to release the plume at $\sim 55\,\mathrm{m}$ height, and the other sources were assumed to be closer to the ground. As we are aware that the uncertainness in wind speed is one key error source for the emission estimate, we made an attempt to at least partly validate these nearsurface winds. For that we compared the 50 m height average calculated from ERA5 over weather station data near the plume locations (see new Sec. A). We found that the wind speed from ERA5 seemed to significantly underestimate the wind speed, with a mean difference of $+2.5\,\mathrm{m\,s^{-1}}$ and single differences up to $+3.4\,\mathrm{m\,s^{-1}}$. Additionally, for the source mapped in P5 the estimated emissions are $\gg 1000\,\mathrm{kg\,hr^{-1}}$ according to other inversions and in situ data (e.g. Jongaramrungruang et al., 2019; Frankenberg et al., 2016). Also in this case the wind speeds from the ground station results in a considerable better match with the published flux estimats. Consequently, we changed our wind speed estimations consistently from the ERA5 50m height average to hourly wind data from ground stations**

nearby and rewrote Sect. 2.2 and 4.3 accordingly, as well as the discussion and conclusions (Sect. 5 and 6). Additionally, we added the new Sect. A to the appendix for the comparison between ERA5 and weather station wind speeds.

R2-6: P5, L4: I couldn't find a description in the text of how the wind direction was assessed from the plume shape. Can you elaborate?

**We visually inspected the plume direction for the best fitting line from the source along the plume. The cross sections were then calculated perpendicular to that. We complemented the according sentences in the manuscript.**

R2-7: Table 1: Can you explain why the PSF-CH4-proxy uncertainty is (slightly) higher than the PSF-CH4 uncertainty? Error stacking from the uncorrelated errors in PSF-CH4 and PSF-CO2?

**Yes, that is what we assumed, too. However, the standard deviation of the fraction is not exactly the root square sum of the single standard deviations and was calculated from the scene separately.**

R2-8: I'm familiar with the CO2 proxy method being applied in the 1650 nm window where there is also strong CO2 absorption, but not in the 2300 nm window. Can you comment on the effectiveness of the method at 2300 nm compared to 1650 nm?

**It works quite well using the strong absorption of $CO_2$ near 2000 nm to retrieve $CO_2$ scaling factors, as can be seen for example in Fig. 7. However, we did not compare the impact of the proxy on the two windows further than what is shown in Table 1.**

R2-9: Can you explain more clearly the pros/cons of WFM-DOAS compared to IMAP-DOAS and Matched Filter? How much faster is WFM than IMAP? When/why is it better than Matched Filter? You say in the abstract that WFM "fills a gap" between the other two retrieval algorithms, and mention some of the differences between them in section 1 and section 5, but I didn't get a clear picture of the motivation behind WFM, except that it's computationally cheaper than IMAP. What are the trade-offs? A concise explanation in the introduction of when/why we should use WFM instead of IMAP or MF would be very helpful.

**We added a quantitative comparison between IMAP-DOAS, MF and WFM-DOAS (see also answer to LG-1). At least from our study we conclude that WFM-DOAS overall produces less background noise. However, in its current form it is not fast enough for near real time analysis of AVIRIS-NG data, which the MF is. The IMAP-DOAS retrieval is more suited for observations deviating larger from the a priori, as through the iterative approach the nonlinear behavior especially of the radiance dependent on gas concentration can be captured better.**

R2-10: P18, L1-10: The double plume structure is intriguing. Could you briefly elaborate on the height of the source, solar zenith angle, and distance between the two plume images in this case? Also, how did you determine wind direction and do the cross-sectional method for this plume?

**We do not know the stack height from direct height measurements or similar. However, from the solar zenith angle during overflight of $\sim 42\,°$ and the length of the shadow of in the RGB image of $\sim 50\,\mathrm{m}$, we used trigonometry to calculate the stack height to be $\sim 55\,\mathrm{m}$. The apparent plume center distance is approximately $50\,\mathrm{m}$. Accounting additionally for the viewing angle (which deviates slightly from nadir), we calculate a plume height of $\sim 56\,\mathrm{m}$ shortly after the beginning of the plumes. Therefore we are confident that this is a real feature. We added the numbers and explained their origin in the manuscript.**

**For the wind direction, we applied the method described in answer R2-6 to the plume structure expanding from the top of the tower, not the top of the shadow. For the cross sectional flux method, one has to have in mind that for the double plume the light passes the plume only once, whereas it is assumed to pass the plume twice in the retrieval and conversion to columns. Summing over the total enhancements of a cross section through both plume structures yields the total enhancement of the plume.**

R2-11: P18, L28: How do you define inside/outside the plume? I think previous AVIRIS studies used a threshold, did you do that too?

**We did not use a threshold, but defined spatial boundaries which were definitely aside the plume. This was done to normalize to the background prior to the inversion (similar to e.g. Krautwurst et al., 2017, Fig. 6).**

R2-12: P21, L1: Is the wind speed calculation referred to here the 50-m average from ERA5, or something else?

**Yes, this was the 50-m average from ERA5. However, as described in our answer to R2-5, we changed to using weather station wind data, as the previously used ERA5 data likely strongly underestimated wind speeds near the surface.**

R2-13 Section 4.3: Are these 1-sigma or 2-sigma errors? Or confidence intervals?

**These are 1-sigma standard deviations. We added this to the text.**

R2-14 P21, L26: I don't understand why the higher wind would cause you to underestimate the emission rate. Shouldn't the cross-sectional method be able to deal with this? Each plume transect gives you an emission estimate, and the higher wind speed will result in higher estimates. I can see how the precision would be lower for a weaker plume, since you might have fewer transects to analyze. But a negative bias would imply that you are missing part of each transect. In that case, couldn't you extend the

transects further from the plume axis? This goes back to my question about defining inside/outside the plume.

**Indeed, it is partly compensated by the higher wind speed. On the other hand, a higher wind speed leads to less molecules in each air parcel. This might drop larger parts of the plume significantly below the background noise or the retrieval detection limit and therefore would not lead to a detectable increase above the normalized background anymore.**

**Bibliography**

Chapman, J. W., Thompson, D. R., Helmlinger, M. C., Bue, B. D., Green, R. O., Eastwood, M. L., Geier, S., Olson-Duvall, W., and Lundeen, S. R.: Spectral and Radiometric Calibration of the Next Generation Airborne Visible Infrared Spectrometer (AVIRIS-NG), Remote Sensing, 11, 2129, https://doi.org/10.3390/rs11182129, 2019.

Frankenberg, C., Thorpe, A. K., Thompson, D. R., Hulley, G., Kort, E. A., Vance, N., Borchardt, J., Krings, T., Gerilowski, K., Sweeney, C., Conley, S., Bue, B. D., Aubrey, A. D., Hook, S., and Green, R. O.: Airborne methane remote measurements reveal heavy-tail flux distribution in Four Corners region, Proceedings of the National Academy of Sciences, https://doi.org/10.1073/pnas.1605617113, URL http://www.pnas.org/content/early/2016/08/10/1605617113.abstract, 2016.

Jongaramrungruang, S., Frankenberg, C., Matheou, G., Thorpe, A. K., Thompson, D. R., Kuai, L., and Duren, R. M.: Towards accurate methane point-source quantification from high-resolution 2-D plume imagery, Atmospheric Measurement Techniques, 12, 6667–6681, https://doi.org/10.5194/amt-12-6667-2019, 2019.

Krautwurst, S., Gerilowski, K., Jonsson, H. H., Thompson, D. R., Kolyer, R. W., Iraci, L. T., Thorpe, A. K., Horstjann, M., Eastwood, M., Leifer, I., Vigil, S. A., Krings, T., Borchardt, J., Buchwitz, M., Fladeland, M. M., Burrows, J. P., and Bovensmann, H.: Methane emissions from a Californian landfill, determined from airborne remote sensing and in situ measurements, Atmospheric Measurement Techniques, 10, 3429–3452, https://doi.org/10.5194/amt-10-3429-2017, 2017.

**Detection and Quantification of $CH_4$ Plumes using the WFM-DOAS retrieval on AVIRIS-NG hyperspectral data**

Jakob Borchardt[1], Konstantin Gerilowski[1], Sven Krautwurst[1], Heinrich Bovensmann[1], Andrew K. Thorpe[2], David R. Thompson[2], Christian Frankenberg[3,2], Charles E. Miller[2], Riley M. Duren[4,2], and John Philip Burrows[1]

[1]Institute of Environmental Physics (IUP), University of Bremen, Bremen, Germany
[2]Jet Propulsion Laboratory, California Institute of Technology, Pasadena, CA, United States of America
[3]California Institute of Technology, Division of Geological and Planetary Sciences, Pasadena, CA, United States of America
[4]Institutes for Resilience, University of Arizona, Tuscon, AZ, United States of America

**Correspondence:** Jakob Borchardt (jakob.borchardt@iup.physik.uni-bremen.de)

**Abstract.** Methane is the second most important anthropogenic greenhouse gas in the Earth's atmosphere.  To effectively reduce these emissions, a good knowledge of source  locations and strengths is required.  Airborne remote sensing instruments such as the Airborne Visible InfraRed Imaging Spectrometer - Next Generation (AVIRIS-NG) with meter scale imaging capabilities are able to yield information about the locations and magnitudes of methane sources

 . In this study we successfully applied the Weighting Function Modified Differential Optical Absorption Spectroscopy (WFM-DOAS) algorithm to AVIRIS-NG  data measured in Canada and the Four Corners region. The WFM-DOAS retrieval is conceptually located between the statistical matched filter (MF)  and the optimal estimation based Iterative Maximum A Posteriori DOAS (IMAP-DOAS) retrieval  algorithm, which already were applied successfully to AVIRIS-NG data  . The WFM-DOAS algorithm is based on a first order Taylor series approximation of the Lambert-Beer law  using only one precalculated radiative transfer calculation  per scene. This yields fast quantitative processing of large data sets.  We detected several methane plumes in the AVIRIS-NG  images recorded during the ABoVE Airborne Campaign and successfully retrieved a coal mine ventilation shaft plume observed during the Four Corners measurement campaign. Comparison between IMAP-DOAS, MF and WFM-DOAS showed good agreement for the coal mine ventilation shaft plume. Additional comparison between MF

and WFM-DOAS for a subset of plumes showed good agreement for one plume and some differences for the other. For five 
[revised manuscript text omitted]

In this study, we test and apply an adaption of the Weighting Function Modified Differential Optical Absorption Spectroscopy (WFM-DOAS) algorithm, used previously for the higher spectral resolution MAMAP measurements (spectral resolution

5 $\sim 0.9\,\mathrm{nm}$ Krings et al., 2011), to hyperspectral AVIRIS-NG data (spectral resolution $\sim 6\,\mathrm{nm}$). The data set was aquired during the Arctic Boreal Vulnerability Experiment Airborne Campaign (ABoVE, Miller et al., 2019b) in Canada and Alaska, which included overflights of multiple coal, oil and gas production sites. The WFM-DOAS approach uses assumptions on the background state of the atmosphere at the time and location of the overflight, including scattering. It performs a linear fit of atmospheric parameters deviating from this background state, making it a fast quantitative method compared to iterative re-

10 trievals. We identified multiple plumes in the retrieval results, and for  five of them the emissions were estimated by application of a cross sectional flux method.

This publication is organized as follows: Following this introduction, Sect. 2 gives an overview of the instrument and data sets. Section 2.1 describes the AVIRIS-NG instrument and radiance data, and Sect. 2.2 introduces the  used meteorological data briefly. In Sect. 3, we present the retrieval algorithm for $CH_4$ and the subsequent filtering. First, we describe the

15 WFM-DOAS method used to infer methane enhancement maps from the spectra in Sect. 3.1. Section 3.2 justifies the fitting windows we use in the retrieval. Sect. 3.3 evaluates the sensitivity of the retrieval to assumptions in the forward model and in Sect. 3.4 we implement a filtering to remove certain error cases. We present experimental results in Sect. 4. First, the detection of plumes is described in Sect. 4.1. Second,  in Sect. 4.2 we compare the WFM-DOAS results with IMAP-DOAS and MF retrieval results. Finally, Sect. 4.3 illustrates a flux inversion using the cross sectional flux method  and

20 shows the results and uncertainties of the emission estimate for  five plumes. The results are discussed in Sect. 5. Sect. 6 summarizes the findings of this study.

**2 Instrument and data sets**

**2.1 The AVIRIS-NG instrument and measurements**

AVIRIS-NG is a hyperspectral imaging spectrometer with a spectral sampling of $\sim 5\,\mathrm{nm}$ and a spectral resolution of $\sim 5-6\,\mathrm{nm}$,

25 depending on the wavelength (Hamlin et al., 2011; Chapman et al., 2019). As a nadir looking instrument, it measures solar radiation reflected from the ground in the wavelength range from $380$ to $2450\,\mathrm{nm}$ with a high signal-to-noise ratio of up to $800$ at $2200\,\mathrm{nm}$ (Thorpe et al., 2016a). The instrument contains 600 spatial pixels, each having a 1 mrad field of view. This results in individual samples with 5 m spatial resolution and a 3 km swath from a typical flight altitude of 5 km above ground level. This allows it to scan large areas in short periods of time. The level-1 data distributed by the operations team

30 contains orthorectified (and gridded) absolute radiances (Chapman et al., 2019), with additional data containing observation parameters such as flight altitude, both solar and instrument zenith and azimuth angles and surface elevation among others (see Miller et al., 2019a, for data description).

For this study, we analyzed a subset of the measurements collected during the ABoVE Airborne Campaign (Miller et al., 2019b) in 2017. The ABoVE campaign aimed to better understand the impacts of environmental changes in Alaska and western Canada. During the airborne campaign, several flight lines of the AVIRIS-NG instrument covered fossil fuel infrastructure in Canada, which contained multiple potential sources for $CH_4$ emission plumes.

The data analyzed in this paper had been preselected to cover a wide range of surface types (e.g. forest, mountainous regions, sand, grass) and at-sensor radiance levels, as well as different flight altitudes. Additionally, the tracks contained different emission sources detected using the matched filter (MF) algorithm (Thompson et al., 2015) to test the retrieval algorithm against known  plume locations over different terrain. The preselection contained 13 flight lines on 5 different days in August 2017, covering different types of sources and surface types. Additionally, to include another strong source under different observation conditions, we included a coal mine ventilation shaft plume observed during the Four Corners measurement campaign in 2015 (Frankenberg et al., 2016).

For the coal mine ventilation shaft plume, already existent MF and IMAP-DOAS retrieval results were used for comparison with the WFM-DOAS results retrieved in the course of this study. Additionally, a subset of the MF results for the flight lines under consideration were utilized for quantitative comparison with WFM-DOAS.

**2.2  Meteorological data from ERA5 and weather stations**

The WFM-DOAS retrieval and the flux inversion require information about various atmospheric parameters in addition to the observed radiances. The following meteorological parameters were extracted from ERA-5 reanalysis data (Copernicus Climate Change Service (C3S) (2017), 2017): hourly data of temperature, pressure and water vapor profiles, as well as height resolved wind speeds and wind components.

For a given flight line, the atmospheric parameters of the nearest 4 spatial grid points and the nearest two time steps of the ERA-5 data set were linearly interpolated to the time and location of the flight line. For the wind speed, we used hourly mean surface wind speed data obtained from nearby weather stations (see also Sect. A for a brief, localized comparison of these data with ERA5 wind speed data). The wind direction for the inversion was estimated from the plume structure itself. For that, we visually inspected the plume direction for the best fitting line from the source along the plume.

[revised manuscript text omitted]

$$CH_{4,\text{enh}} = \left( \frac{PSF_{CH_4}}{PSF_{CO_2}} \middle/ PSF_{\text{proxy, bg}} - 1 \right) \cdot CH_{4,\text{back}} \cdot k_{\text{AK}}. \tag{2}$$

A discussion of the biases introduced by the assumptions made for the WFM-DOAS retrieval are studied in the sensitivity analysis in Sect. 3.3.

**3.2 Comparison of major fitting windows in the SWIR spectral range**

For the MAMAP instrument the fitting window was $\sim 1630\,\text{nm} - 1675\,\text{nm}$ for $CH_4$ and $\sim 1592\,\text{nm} - 1617\,\text{nm}$ for $CO_2$ at $0.9\,\text{nm}$ spectral resolution due to the sensor design (see Krings et al., 2011; Gerilowski et al., 2011). AVIRIS-NG additionally offered the possibility to fit the $CH_4$ and $CO_2$ absorption lines between $2000\,\text{nm}$ and $2400\,\text{nm}$ for the retrieval of $CH_4$, although at a coarser spectral resolution of $5.5\,\text{nm} - 6.0\,\text{nm}$. For example, the IMAP-DOAS retrieval successfully retrieved $CH_4$ concentrations from AVIRIS-NG data using the spectral regions of $2215\,\text{nm} - 2410\,\text{nm}$ for $CH_4$ and $1904\,\text{nm} - 2099\,\text{nm}$ for $CO_2$ (Thorpe et al., 2017).

For a first assessment of those absorption bands, we convolved a simulated high resolution spectrum and the corresponding weighting functions for $CH_4$, $CO_2$ and $H_2O$ with the AVIRIS-NG instrument spectral response function. We used a Gaussian spectral response, where the FWHM were distributed as part of the data set. Fig. 2 shows the results of convolution and resampling to the AVIRIS-NG wavelength grid.

[revised manuscript text omitted]

5    then perturbed these atmospheric parameters to create synthetic AVIRIS-NG observations at instrument spectral resolution. Next, we applied the WFM-DOAS algorithm to these simulated measurements, and assessed the systematic offset from the expected PSF value for $PSF_{CH_4}$, $PSF_{CO_2}$ and $PSF_{CH_4,proxy}$. To assess the influence of linearization on the retrieval results, we did not include instrument noise in this analysis. The background simulation was based on the parameters extracted for one flight line observed with a nadir viewing angle. The $CH_4$ enhancements and a plume from this flight line are shown in Fig. 8.

10    For the sensitivity analysis, we perturbed the following set of parameters (Table 2): the aircraft altitude, the surface elevation, the instrument viewing angle (i.e. the instrument zenith angle) and the surface albedo as geometric parameters; and the total columns of $CH_4$, $CO_2$ and $H_2O$, and the pressure and temperature profiles as atmospheric parameters. Additionally, we used selected spectral reflectance spectra of different surfaces instead of a spectrally uniform albedo and examined two additional aerosol scenarios. We did not analyze the sensitivity to the solar zenith and azimuth angles, since these angles were effectively

15    constant over the timespan of a flight line. In addition, we did not analyze the instrument azimuth angle dependency since the flight tracks were nearly straight and the azimuth angle therefore effectively constant for a flight line. Finally, we did not evaluate sensitivity to spectral response function and wavelength calibration, since these were adjusted during conversion of the raw digital numbers to radiometric data cubes (Chapman et al., 2019).

[revised manuscript text omitted]

20   extend over longer ranges and have unambiguous morphology in Fig. 8 and Fig. 9. The additional $CH_4$ plumes and enhancements  can be found in the appendix in Sect. C. Those comprise, among others, emissions most likely resulting from open cast coal mining (Fig. C1) or a well pad located in a forest (Fig. C2).

In Fig. 8, two overpasses of the same source on two different days are shown. On the first day (Fig. 8 (b)), the plume structure

25  was recognizable over a relatively long distance, while on the second day (Fig. 8 (d)), the plume was only faintly visible in the vicinity of the source. This was most likely due to the wind speed, which was significantly higher on the second day ($\sim$7.6 m s$^{-1}$) compared to the first day ( 3.7 m s$^{-1}$). Assuming a constant emission, a wind speed higher by a factor of  roughly 2 means a decrease in the column enhancements by a factor of  roughly 2, which would significantly reduce the visibility of the plume in the retrieval results by diluting larger parts of the plume faster below

30  the background noise.

In Fig. 8, a new interesting feature was observable at the source. There, we observed a double plume structure that was especially prominent during the first overpass. Comparison with the RGB image revealed, that one part seemed to originate from the vent, while the other part seemed to originate from the top of the shadow of the vent. The vent released the plume

several meters above the surface. Because the plume was very narrow near the source, the sunlight only passed the plume either before or after hitting the ground. As those two light paths were attributed to different ground scenes, the absorption and therefore the apparent $CH_4$ enhancements were visible at two locations leading to the double plume structure. Further down the plume, atmospheric mixing took place and the plume widened. Then, the sunlight passed through the plume both before and after hitting the ground, and the double plume structure vanished. A simple geometric consideration of the distance between the two plume structures ($\sim 50\,\mathrm{m}$), the solar and instrument zenith  angles ($\sim 42\,°$and $\sim 2.5\,°$), and the vent height  calculated from the shadow of the structure ($\sim 55\,\mathrm{m}$) and the solar zenith angle supported this hypothesis.

In Fig. 9, two plumes originating from well pads are shown. Both extended linearly from their source and were visible over approximately $100\,\mathrm{m}$. While the first plume (Fig. 9, b) originated from a cold vent (similar to Fig. 8), the emitting structure for the plume in Fig. 9 (d) could not be identified from the RGB images. It seemed, however, that the source was located near the surface. This would also explain the large deviation of the plume direction from the wind direction aquired from the ERA-5 model data, since the nearby forests could have significantly altered the wind direction near the surface.

Additionally, the plume detected over a coal mine ventilation shaft during the Four Corners measurement campaign in 2015 is shown in Fig. 10. The plume was nearly one kilometer long, with a straight profile except for a small diversion at the tip that suggested a change in wind direction.

**4.2 Comparison of WFM-DOAS retrieval results with IMAP-DOAS and MF results**

To assess the performance of the WFM-DOAS retrieval with respect to the IMAP-DOAS and MF retrieval results, a subset of the data was compared quantitatively. We focused here on the coal mine ventilation shaft plume for all three retrievals, and additionally compared the WFM-DOAS results to MF results over the plumes P1 - P4. As IMAP-DOAS and the MF retrieve $CH_4$ enhancements below the aircraft in $\mathrm{ppm} \cdot \mathrm{m}$ ($CH_{4,\mathrm{enh,ppm\,m}}$), we converted these measurements to enhancements in $\mathrm{molec\,cm^{-2}}$ ($CH_{4,\mathrm{enh,MF}}$) via the following equation:

$$CH_{4,\mathrm{enh,MF/IMAP}} = \frac{CH_{4,\mathrm{enh,ppm\,m}}}{h_{\mathrm{airc}}} \cdot \mathrm{subcol}_{\mathrm{tot}} \cdot 10^{-6}, \tag{3}$$

where $h_{\mathrm{airc}}$ is the distance between instrument and ground in nadir direction, and $\mathrm{subcol}_{\mathrm{tot}}$ is the number of dry molecules per square centimeter between aircraft and ground. The sub column below the aircraft was calculated on the basis of the profiles used for the WFM-DOAS retrieval.

We compared the retrieval results in two different ways, focusing on retrieval scatter and retrieved enhancements. First, for the retrieval scatter, we calculated the standard error (1-sigma) of retrieved enhancements over areas near the plumes P1 - P5. This is summarized in Table 5. For all plume regions except P5 the retrieval scatter is lower for WFM-DOAS than for the MF results, which indicate less retrieval noise. For P5, where all three retrieval results were available, the background noise is very similar for all retrieval methods.

Second, to evaluate the agreement of the retrieval results, we additionally produced scatter plots of the central plume area. We added the 1:1 line and a linear fit to the scatter plots. For the coal mine ventilation shaft plume all three retrievals agreed well

**Table 5.** Comparison of background noise of the retrievals based on the standard deviation of retrieval results near the plumes. As we only have IMAP-DOAS results for one plume, the other fields are left empty.

|  | P1 (molec cm$^{-2}$) | P2 (molec cm$^{-2}$) | P3 (molec cm$^{-2}$) | P4 (molec cm$^{-2}$) | P5 (molec cm$^{-2}$) |
|---|---|---|---|---|---|
| WFM-DOAS | $\pm 1.3 \cdot 10^{17}$ | $\pm 8.3 \cdot 10^{17}$ | $\pm 7.8 \cdot 10^{17}$ | $\pm 1.8 \cdot 10^{18}$ | $\pm 5.2 \cdot 10^{17}$ |
| MF | $\pm 2.0 \cdot 10^{18}$ | $\pm 2.1 \cdot 10^{18}$ | $\pm 1.7 \cdot 10^{18}$ | $\pm 2.4 \cdot 10^{18}$ | $\pm 4.1 \cdot 10^{17}$ |
| IMAP-DOAS |  |  |  |  | $\pm 5.5 \cdot 10^{17}$ |

in the retrieved enhancements, as can be seen in Fig. 11. Compared to the IMAP-DOAS retrieval, WFM-DOAS did retrieve slightly lower enhancements, while it did estimate slightly higher enhancements than MF.

Additionally, we compared the WFM-DOAS and MF results for the plumes P1 - P4 (see Fig. D1). For P3 especially the higher enhancements located inside the plume agree well. However, for the other plumes there is quite some mismatch between MF results and WFM-DOAS results. The cause for the disagreement is currently under investigation and might be related to the use of a standard gas target for the matched filter rather than calculating a target specifically parameterized for a given scene.

**4.3  Flux and uncertainty estimation based on cross sectional flux method**

[revised manuscript text omitted]

**Appendix A:  Comparison of weather station wind data with ERA5 50 m height average data**

We compared the average wind speed over the lowest $50\,\mathrm{m}$ height average of ERA5 data to the hourly mean wind speed data obtained from weather stations near the sources. For P1 and P2, we used wind speed data obtained at the Firebag weather station (Wood Buffalo Environmental Association, 2020) which was located slightly below $20\,\mathrm{km}$ away from the plumes. For P3 and P4, we used wind speed data from the weather stations "Sundre A" and "Patricia AGCM"[1], located $5\,\mathrm{km}$ and $17\,\mathrm{km}$ away from the source. For P5, we used wind speed data from the Four Corners Regional Airport weather station from the MesoWest network (Horel et al., 01 Feb. 2002) $\sim 15\,\mathrm{km}$ away from the plume. A comparison between those ground stations and the average wind speed over the lowest $50\,\mathrm{m}$ of ERA5 data is shown in Fig. A1.
* * *
[1]Data provided by Alberta Agriculture and Forestry, Alberta Climate Information Service (ACIS) https://acis.alberta.ca (retrieved in Dec. 2020)

As is clearly visible, the ERA5 data significantly deviated from the wind speeds present at the weather stations. The mean deviation was $+2.5\,\mathrm{m\,s^{-1}}$, with single differences up to $+3.4\,\mathrm{m\,s^{-1}}$. Consequently, we used the weather station wind data for further analysis in the flux inversion.

**Appendix B: Striping Effect**

[revised manuscript text omitted]

**Flight line ang20150422t164506 - Source P5**

(a) RGB Image                    (b) WFM-DOAS CH$_4$ enhancement

[Figure]

CH$_4$ enhancement in molec / cm$^2$

**Figure 10.** Similar to Fig. 8, but for the exhaust plume of a coal mine ventilation shaft in the Four Corners region.

[Figure]

**Figure 11.** Scatter plot comparison between IMAP-DOAS and WFM-DOAS (a) and MF and IMAP-DOAS (b) respectively for the coal mine ventilation shaft plume. For the red line only enhancements above the background noise stated in Table 5 were analyzed, which should be largely dominated by the plume.

[Figure]

**Figure A1.** Comparison between wind speed averaged over the lowest 50 m above ground for ERA5 data and collocated wind station data. ERA5 data significantly underestimated the wind speed present at a given time.

[Figure]

**Figure C1.**  Same as Fig. 9, but for methane enhancements after application of the proxy resulting from bituminous coal extraction.  The upper two measurements  were taken within ∼ 10 minutes, and show the  same brim. The dashed line is plotted on the same location in both images. While in (b) a plume emanating from the brim is visible, in (d) strong accumulations near the brim are visible. In (f) the plume is more diffuse and the highest enhancements are  visible near a brim. However, it is not as clear as in (b) and (d), if it really originates from the brim or might be an artifact.

[Figure]

**Figure C2.** Similar to Fig. C1, but for methane enhancements from oil/gas infrastructure. In (a) and (b), the emissions originate from a well pad in a forest. Due to the low radiance over the trees, only the well pad itself passes the quality filters. The enhancements seem accumulated and no clear wind direction is visible. In (c) and (d), a facility located at a bitumen extraction site is shown and meanders through the facility.

[Figure]

**Figure C3.** Similar to Fig. C1, but for two overpasses over cold vents at a bitumen extraction site. The plumes are only faint, especially for the second overpass (d). In the results a small striping effect is still visible. The destriping reduces the effect, but is not able to totally eliminate it.

**Appendix D:  Additional comparison between MF and WFM-DOAS**

In addition to the comparison of IMAP-DOAS and MF for the same plume, we compared MF and WFM-DOAS retrieval results for subregions zoomed in on the plumes P1 - P4. For the scene P3 the retrievals seem to agree quite well. Especially the higher enhancements correlated with the plume lay near the 1:1 line. However, for the plumes P1, P2 and P4, there is a bigger discrepancy between MF and WFM-DOAS, which can not be explained by the larger scatter alone. The cause for this is currently under investigation.

[Figure]

**Figure D1.** Similar to Fig. 11, but for the plume regions P1 - P4.

*Author contributions.* JB contributed to the study design, adapted the retrieval, analyzed the data and wrote the manuscript, KG and HB initialized the study, KG and SK contributed to the retrieval adaption, study design and paper draft, AKT, DRT and CF helped in data set selection and with the handling of the data, RD and CM designed the flight plans, organized and led the flights, HB and JPB supervised the study, contributing to the scientific objectives and interpretation of results. All authors contributed to the final manuscript.

5    *Competing interests.* The authors declare that they have no conflict of interest.

*Acknowledgements.* This work was funded in parts by the BMBF project AIRSPACE MAMAP2D (FKZ01LK1701B). We would also like to acknowledge the contributions of the AVIRIS flight and instrument teams. The AVIRIS-NG data were collected as part of the Arctic Boreal Vulnerability Experiment (ABoVE), a NASA Terrestrial Ecology program, and other methane studies supported by NASA's Earth Science Division. A portion of this work was performed at the Jet Propulsion Laboratory, California Institute of Technology, under contract

10    with NASA. The hourly resolved wind data for P3 and P4 were provided provided by Alberta Agriculture and Forestry, Alberta Climate Information Service (ACIS), https://acis.alberta.ca for stations Patricia AGCM and Sundre A (Data retrieved on Dec, 2020)

---

## Author Comment (AC2)

**Author Response to Luis Guanter (LG)**

**First, we want to thank Luis Guanter for the careful and concise review and the helpful comments. In the following, we will first repeat the review comment (LG-X), and then give our response.**

LG-1: Comparison to Matched Fiter (MF) and IMAP-DOAS methods: according to the authors, the motivation to propose a WFM-DOAS retrieval for AVIRIS-NG is to "fill a gap" between MF and IMAP-DOAS approaches. However, nothing is said about the retrieval performance of the proposed WFM-DOAS with respect to the other two methods. I believe that the study would greatly benefit from a quantitative comparison of the three retrievals. The MF XCH4 data for the same ABOVE data set seem to be already available (p4, L20), and looking at the author list I understand that it wouldn't be difficult to also have IMAP-DOAS retrievals for at least a data subset containing one of the identified plumes.

**We agree that this comparison between all three methods would greatly enhance the paper. Consequently, we included one coal mine ventilation shaft plume observed in the Four Corners measurement campaign, for which also already IMAP-DOAS and MF results existed, and retrieved this plume with the WFM-DOAS method presented in this paper for comparison of all three methods. Additionally, we compared the MF and WFM-DOAS results for the plumes which were analyzed quantitatively in the main text in the new Section 4.3.**

LG-2: Selection of the AVIRIS-NG data set used for this: the authors don't provide any explanation of why they select this particular ABOVE AVIRIS-NG data set for the study, but I would say that there could be better AVIRIS / AVIRIS-NG data sets for a study focused on the presentation of a new retrieval method. For example, the data sets from the different campaigns that JPL has run over the last years, including e.g. the one over Four Corners (Frankenberg et al., 2016) and California (Duren et al., 2019), for which the L1B data are surely available, could offer better observation conditions, properly documented plumes and fluxes, and a wider range of emission intensities and types. So I would recommend the authors to extend the study to other AVIRIS-NG showing other acquisition conditions, and in particular larger emissions. On the other hand, if the authors were interested in the particularly challenging conditions of the Canada sites, this should be discussed in the text.

**We chose the ABoVE data as it provided a wide range of surface types and shapes and observation geometries to test the retrieval under different, partly challenging circumstances. We added the explanation for the selection of this data set to Section 2.1. To include a source with higher enhancements of $CH_4$, we added a flight line over a coal mine ventilation shaft observed during the Four Corners measurement campaign.**

LG-3: Abstract: it might benefit from shortening and removal of line breaks

**We agree and shortened the abstract.**

LG-4: p4 L25: "Copernicus", nested brackets

**Done**

LG-5: Fig. 1 caption: "can not be fully resolved"

**Done**

LG-6: p7, L28: plus surface reflectance shows in general less variability at 1600 nm than at 2300 nm

**The mean reflectance for different surface types varies similarly for 1600 nm and 2300 nm. Also some interfering absorbers have spectral features as large around 1650 nm as around 2300 nm (See for example Foote et al., 2020, Fig. 2). So we do not think that this is a major advantage of the 1600 nm spectral region.**

LG-7: Section 3.3., sensitivity analysis: it could also be relevant to include uncertainties in the spectral calibration (position and shape of the spectral response function), if this is not optimized in the retrieval

**The spectral calibration is optimized during the L0 $\rightarrow$ L1b processing (see Chapman et al., 2019). Nevertheless, we fitted additional wavelength shift and squeeze parameters in the retrieval. We added the explanation to Sect. 3.3.**

LG-8: p15, L13 "determined by the following"

**Done**

LG-9: p17, L9 – the cross sectional method hasn't been mentioned until now

**We changed the sentence in the following way to address this criticism: "However, many plumes were faint or located near infrastructure, making unambiguous detection difficult or the plumes were very short. Therefore, we show plumes which extend over longer ranges and have unambiguous morphology in Fig. 8 and Fig. 9. The additional $CH_4$ plumes and enhancements can be found in the appendix in Sect. B."**

LG-10: p17, L17: "A wind speed higher by a factor 1.6 means..."? why is this? can you please explain this further?

**Assuming a constant flux, if the wind speed is higher by a constant factor, the total column enhancements along the cross section have to be lower by that factor. However, a reduction in total column enhancements is less visible above the background and dilutes faster. This is only true if we assume a constant emitting source, which indeed might not be the case here. Due to the wind speed uncertainties however we can not dismiss this possibility.**

LG-11: Section 4.2 and 4.3 could be merged. In general, a manuscript structure with separate Methods and Results sections would be better.

**We decided against such a structure, as the selection and description of some methods relied on the results from previous parts. However, merging Section 4.2 and 4.3 is indeed more logical in the current structure.**

LG-12: Figure captions: perhaps just a matter of taste, but I think they could be shorter and avoid the explanations already included in the main text (e.g. Fig. 8 and 9).

**We agree and shortened excessive interpretation in the figures.**

LG-13: Fig. 9, Plume P4: at least from visual inspection, it is not obvious to me that that is a real plume. It could also be the effect of background reflectance variations. Would it make sense to show a map of at-sensor radiance at 2200 nm and its slope to discard this possibility?

**We produced this plot for comparison between the radiance at $2140\,\mathrm{nm}$ and the plume. The structures visible in the gray scale image of radiance at $2140\,\mathrm{nm}$ do not seem to match the faint plume structure. Consequently, we believe this is a real $CH_4$ plume.**

[Figure]

Figure 0.1: Plume P4 (b) and Radiance at $2140\,\mathrm{nm}$ (a) to check for the possibility of surface effects being mistaken for a plume structure.

LG-14: Refs. At least Cusworth et al. (https://doi.org/10.5194/amt-12-5655-2019) and Foote et al. (https://doi.org/10.1109/TGRS.2020.2976888) should be added

**We agree and added the citations.**

**Bibliography**

Chapman, J. W., Thompson, D. R., Helmlinger, M. C., Bue, B. D., Green, R. O., Eastwood, M. L., Geier, S., Olson-Duvall, W., and Lundeen, S. R.: Spectral and Radiometric Calibration of the Next Generation Airborne Visible Infrared Spectrometer (AVIRIS-NG), Remote Sensing, 11, 2129, https://doi.org/10.3390/rs11182129, 2019.

Foote, M. D., Dennison, P. E., Thorpe, A. K., Thompson, D. R., Jongaramrungruang, S., Frankenberg, C., and Joshi, S. C.: Fast and Accurate Retrieval of Methane Concentration From Imaging Spectrometer Data Using Sparsity Prior, IEEE Transactions on Geoscience and Remote Sensing, 58, 6480–6492, https://doi.org/10.1109/tgrs.2020.2976888, 2020.

**Detection and Quantification of $CH_4$ Plumes using the WFM-DOAS retrieval on AVIRIS-NG hyperspectral data**

Jakob Borchardt[1], Konstantin Gerilowski[1], Sven Krautwurst[1], Heinrich Bovensmann[1], Andrew K. Thorpe[2], David R. Thompson[2], Christian Frankenberg[3,2], Charles E. Miller[2], Riley M. Duren[4,2], and John Philip Burrows[1]

[1]Institute of Environmental Physics (IUP), University of Bremen, Bremen, Germany
[2]Jet Propulsion Laboratory, California Institute of Technology, Pasadena, CA, United States of America
[3]California Institute of Technology, Division of Geological and Planetary Sciences, Pasadena, CA, United States of America
[4]Institutes for Resilience, University of Arizona, Tuscon, AZ, United States of America

**Correspondence:** Jakob Borchardt (jakob.borchardt@iup.physik.uni-bremen.de)

**Abstract.** Methane is the second most important anthropogenic greenhouse gas in the Earth's atmosphere.  To effectively reduce these emissions, a good knowledge of source  locations and strengths is required.  Airborne remote sensing instruments such as the Airborne Visible InfraRed Imaging Spectrometer - Next Generation (AVIRIS-NG) with meter scale imaging capabilities are able to yield information about the locations and magnitudes of methane sources

. In this study we successfully applied the Weighting Function Modified Differential Optical Absorption Spectroscopy (WFM-DOAS) algorithm to AVIRIS-NG  data measured in Canada and the Four Corners region. The WFM-DOAS retrieval is conceptually located between the statistical matched filter (MF)  and the optimal estimation based Iterative Maximum A Posteriori DOAS (IMAP-DOAS) retrieval  algorithm, which already were applied successfully to AVIRIS-NG data. The WFM-DOAS algorithm is based on a first order Taylor series approximation of the Lambert-Beer law  using only one precalculated radiative transfer calculation  per scene. This yields fast quantitative processing of large data sets.  We detected several methane plumes in the AVIRIS-NG  images recorded during the ABoVE Airborne Campaign and successfully retrieved a coal mine ventilation shaft plume observed during the Four Corners measurement campaign. Comparison between IMAP-DOAS, MF and WFM-DOAS showed good agreement for the coal mine ventilation shaft plume. Additional comparison between MF

and WFM-DOAS for a subset of plumes showed good agreement for one plume and some differences for the other. For five plumes the emissions were estimated using a simple cross sectional flux method. The retrieved fluxes originated from well pads, cold vents and a coal mine ventilation shaft and ranged between $(155 \pm 71)\,\mathrm{kg\,(CH_4)\,h^{-1}}$ and $(1220 \pm 450)\,\mathrm{kg\,(CH_4)\,h^{-1}}$. The wind velocity 
[revised manuscript text omitted]

In this study, we test and apply an adaption of the Weighting Function Modified Differential Optical Absorption Spectroscopy (WFM-DOAS) algorithm, used previously for the higher spectral resolution MAMAP measurements (spectral resolution $\sim 0.9\,\mathrm{nm}$ Krings et al., 2011), to hyperspectral AVIRIS-NG data (spectral resolution $\sim 6\,\mathrm{nm}$). The data set was aquired during the Arctic Boreal Vulnerability Experiment Airborne Campaign (ABoVE, Miller et al., 2019b) in Canada and Alaska, which included overflights of multiple coal, oil and gas production sites. The WFM-DOAS approach uses assumptions on the background state of the atmosphere at the time and location of the overflight, including scattering. It performs a linear fit of atmospheric parameters deviating from this background state, making it a fast quantitative method compared to iterative retrievals. We identified multiple plumes in the retrieval results, and for  five of them the emissions were estimated by application of a cross sectional flux method.

This publication is organized as follows: Following this introduction, Sect. 2 gives an overview of the instrument and data sets. Section 2.1 describes the AVIRIS-NG instrument and radiance data, and Sect. 2.2 introduces the  used meteorological data briefly. In Sect. 3, we present the retrieval algorithm for $CH_4$ and the subsequent filtering. First, we describe the WFM-DOAS method used to infer methane enhancement maps from the spectra in Sect. 3.1. Section 3.2 justifies the fitting windows we use in the retrieval. Sect. 3.3 evaluates the sensitivity of the retrieval to assumptions in the forward model and in Sect. 3.4 we implement a filtering to remove certain error cases. We present experimental results in Sect. 4. First, the detection of plumes is described in Sect. 4.1. Second,  in Sect. 4.2 we compare the WFM-DOAS results with IMAP-DOAS and MF retrieval results. Finally, Sect. 4.3 illustrates a flux inversion using the cross sectional flux method  and shows the results and uncertainties of the emission estimate for  five plumes. The results are discussed in Sect. 5. Sect. 6 summarizes the findings of this study.

**2 Instrument and data sets**

**2.1 The AVIRIS-NG instrument and measurements**

AVIRIS-NG is a hyperspectral imaging spectrometer with a spectral sampling of $\sim 5\,\mathrm{nm}$ and a spectral resolution of $\sim 5-6\,\mathrm{nm}$, depending on the wavelength (Hamlin et al., 2011; Chapman et al., 2019). As a nadir looking instrument, it measures solar radiation reflected from the ground in the wavelength range from $380$ to $2450\,\mathrm{nm}$ with a high signal-to-noise ratio of up to $800$ at $2200\,\mathrm{nm}$ (Thorpe et al., 2016a). The instrument contains $600$ spatial pixels, each having a 1 mrad field of view. This results in individual samples with 5 m spatial resolution and a 3 km swath from a typical flight altitude of 5 km above ground level. This allows it to scan large areas in short periods of time. The level-1 data distributed by the operations team contains orthorectified (and gridded) absolute radiances (Chapman et al., 2019), with additional data containing observation parameters such as flight altitude, both solar and instrument zenith and azimuth angles and surface elevation among others (see Miller et al., 2019a, for data description).

For this study, we analyzed a subset of the measurements collected during the ABoVE Airborne Campaign (Miller et al., 2019b) in 2017. The ABoVE campaign aimed to better understand the impacts of environmental changes in Alaska and western Canada. During the airborne campaign, several flight lines of the AVIRIS-NG instrument covered fossil fuel infrastructure in Canada, which contained multiple potential sources for $CH_4$ emission plumes.

The data analyzed in this paper had been preselected to cover a wide range of surface types (e.g. forest, mountainous regions, sand, grass) and at-sensor radiance levels, as well as different flight altitudes. Additionally, the tracks contained different emission sources detected using the matched filter (MF) algorithm (Thompson et al., 2015) to test the retrieval algorithm against known  plume locations over different terrain. The preselection contained 13 flight lines on 5 different days in August 2017, covering different types of sources and surface types. Additionally, to include another strong source under different observation conditions, we included a coal mine ventilation shaft plume observed during the Four Corners measurement campaign in 2015 (Frankenberg et al., 2016).

For the coal mine ventilation shaft plume, already existent MF and IMAP-DOAS retrieval results were used for comparison with the WFM-DOAS results retrieved in the course of this study. Additionally, a subset of the MF results for the flight lines under consideration were utilized for quantitative comparison with WFM-DOAS.

**2.2  Meteorological data from ERA5 and weather stations**

The WFM-DOAS retrieval and the flux inversion require information about various atmospheric parameters in addition to the observed radiances. The following meteorological parameters were extracted from ERA-5 reanalysis data (Copernicus Climate Change Service (C3S) (2017), 2017): hourly data of temperature, pressure and water vapor profiles, as well as height resolved wind speeds and wind components.

For a given flight line, the atmospheric parameters of the nearest 4 spatial grid points and the nearest two time steps of the ERA-5 data set were linearly interpolated to the time and location of the flight line. For the wind speed, we used hourly mean surface wind speed data obtained from nearby weather stations (see also Sect. A for a brief, localized comparison of these data with ERA5 wind speed data). The wind direction for the inversion was estimated from the plume structure itself. For that, we visually inspected the plume direction for the best fitting line from the source along the plume.

[revised manuscript text omitted]
 1630\,\text{nm} - 1675\,\text{nm}$ for $CH_4$ and $\sim 1592\,\text{nm} - 1617\,\text{nm}$ for $CO_2$ at $0.9\,\text{nm}$ spectral resolution due to the sensor design (see Krings et al., 2011; Gerilowski et al., 2011). AVIRIS-NG additionally offered the possibility to fit the $CH_4$ and $CO_2$ absorption lines between $2000\,\text{nm}$ and $2400\,\text{nm}$ for the retrieval of $CH_4$, although at a coarser spectral resolution of $5.5\,\text{nm} - 6.0\,\text{nm}$. For example, the IMAP-DOAS retrieval successfully retrieved $CH_4$ concentrations from AVIRIS-NG data using the spectral regions of $2215\,\text{nm} - 2410\,\text{nm}$ for $CH_4$ and $1904\,\text{nm} - 2099\,\text{nm}$ for $CO_2$ (Thorpe et al., 2017).

For a first assessment of those absorption bands, we convolved a simulated high resolution spectrum and the corresponding weighting functions for $CH_4$, $CO_2$ and $H_2O$ with the AVIRIS-NG instrument spectral response function. We used a Gaussian spectral response, where the FWHM were distributed as part of the data set. Fig. 2 shows the results of convolution and resampling to the AVIRIS-NG wavelength grid.

Both fitting windows had their advantages and disadvantages, especially for the lower spectral resolution of AVIRIS-NG. Around $2300\,\text{nm}$, the absorption features of $CH_4$ were about a factor of 2 stronger and had a more pronounced structure, which could lead to a better detection of methane changes. Around $1650\,\text{nm}$, the at sensor radiance was nearly twice as high for the same albedo, which could mean a higher signal-to-noise ratio. Additionally, there was less overlap with water vapor absorption features near $1600\,\text{nm}$.

We used a two step approach to find the best fitting window: First, we created a spatially averaged spectrum over a homogeneous surface elevation and surface type to reduce the instrument noise and systematic influences. Then, we optimized the edges of both fit windows for fitting the gas features in each window. As a measure of fit quality, the root mean square error (RMSE) between measurement and fit result was used. For $CH_4$, the best fitting windows were $1625\,\text{nm} - 1700\,\text{nm}$ and $2235\,\text{nm} - 2380\,\text{nm}$, and for $CO_2$ $1550\,\text{nm} - 1620\,\text{nm}$ and $2040\,\text{nm} - 2100\,\text{nm}$ (see also Fig. 2). For simplicity, the fitting windows between $1550\,\text{nm}$ and $1700\,\text{nm}$ will be called "weak windows", and the fitting windows between $2040\,\text{nm}$ and $2380\,\text{nm}$ will be called "strong windows" in the following parts, according to the depth of the absorption features.

To assess the measurement precision in each window, we selected a homogeneous, flat, bright area which contained no potential sources. We then applied the retrieval to the whole flight line containing this test case for each of the fitting windows and gases. These initial results showed detector column dependent stripes (see Sect. B in the appendix). To correct this effect, we normalized the $PSF_{CH_4}$, $PSF_{CO_2}$ and $PSF_{CH_4,\text{proxy}}$ for each pixel by the median PSF of its corresponding detector column. We selected the median for resilience against outliers, which could otherwise have a large impact on the correction

**Spectrum and weighting functions**

[Figure]

**Figure 2.** The high resolution simulated spectra (in (a) and (b) green) are convolved with the slit function of AVIRIS-NG and sampled to the AVIRIS-NG wavelength grid (solid black line in upper row). The lower row ((c) and (d)) shows the weighting functions, i.e. the change of intensity due to a change in atmospheric concentration for $CH_4$ (blue), $CO_2$ (black) and $H_2O$ (red). The shaded areas denote the fitting windows for $CO_2$ (gray) and $CH_4$ (light orange).

After destriping, we compared the standard deviation in the test case region of the weak and strong window retrieval results of $PSF_{CH_4}$, $PSF_{CO_2}$ and $PSF_{CH_4,proxy}$ (Table 1. The retrieved $PSF_{CH_4}$ and $PSF_{CH_4,proxy}$ were noisier in the weak window by a factor of 3.3 and 2.9, respectively. The retrieved $PSF_{CO_2}$ was noisier by a factor of 1.5 in the weak window. Therefore, we only used the strong windows in later analyses.

**Table 1.** Comparison of the standard deviation of $PSF_{CH_4}$, $PSF_{CO_4}$ and $PSF_{CH_4,proxy}$ in the two fitting windows around $1645\,nm$ and $2300\,nm$ for the AVIRIS-NG FWHM ($\approx 6\,nm$). The standard deviation was calculated over a homogeneous and flat area with no visible plume and possible source inside. The statistical uncertainties for $PSF_{CH_4}$ and $PSF_{CO_2}$ are therefore uncorrelated.

|  | Standard deviation PSF $1645\,nm$ fitting window | Standard deviation PSF $2300\,nm$ fitting window |
|---|---|---|
| $PSF_{CH_4}$ | $\pm 6.4\%$ | $\pm 1.9\%$ |
| $PSF_{CO_2}$ | $\pm 1.9\%$ | $\pm 1.3\%$ |
| $PSF_{CH_4,proxy}$ | $\pm 6.6\%$ | $\pm 2.3\%$ |

**3.3 Sensitivity analysis**

In addition to the noise in the spectra, uncertainties and variability in the assumed constant atmospheric background parameters could lead to errors in the retrieval results. To assess the magnitude and influence of these deviations, we performed multiple sensitivity analyses. We used a common set of geometric and atmospheric parameters to model the background spectrum. We then perturbed these atmospheric parameters to create synthetic AVIRIS-NG observations at instrument spectral resolution. Next, we applied the WFM-DOAS algorithm to these simulated measurements, and assessed the systematic offset from the expected PSF value for $PSF_{CH_4}$, $PSF_{CO_2}$ and $PSF_{CH_4,proxy}$. To assess the influence of linearization on the retrieval results, we did not include instrument noise in this analysis. The background simulation was based on the parameters extracted for one flight line observed with a nadir viewing angle. The CH$_4$ enhancements and a plume from this flight line are shown in Fig. 8.

For the sensitivity analysis, we perturbed the following set of parameters (Table 2): the aircraft altitude, the surface elevation, the instrument viewing angle (i.e. the instrument zenith angle) and the surface albedo as geometric parameters; and the total columns of CH$_4$, CO$_2$ and H$_2$O, and the pressure and temperature profiles as atmospheric parameters. Additionally, we used selected spectral reflectance spectra of different surfaces instead of a spectrally uniform albedo and examined two additional aerosol scenarios. We did not analyze the sensitivity to the solar zenith and azimuth angles, since these angles were effectively constant over the timespan of a flight line. In addition, we did not analyze the instrument azimuth angle dependency since the flight tracks were nearly straight and the azimuth angle therefore effectively constant for a flight line. Finally, we did not evaluate sensitivity to spectral response function and wavelength calibration, since these were adjusted during conversion of the raw digital numbers to radiometric data cubes (Chapman et al., 2019).

[revised manuscript text omitted]
  or the plumes were very short. Therefore, we show plumes which  extend over longer ranges and have unambiguous morphology in Fig. 8 and Fig. 9. The additional $CH_4$ plumes and enhancements  can be found in the appendix in Sect. C. Those comprise, among others, emissions most likely resulting from open cast coal mining (Fig. C1) or a well pad located in a forest (Fig. C2).

In Fig. 8, two overpasses of the same source on two different days are shown. On the first day (Fig. 8 (b)), the plume structure was recognizable over a relatively long distance, while on the second day (Fig. 8 (d)), the plume was only faintly visible in the vicinity of the source. This was most likely due to the wind speed, which was significantly higher on the second day ($\sim\!4.4\,\mathrm{m\,s^{-1}}$ $7.6\,\mathrm{m\,s^{-1}}$) compared to the first day ($\sim\!2.6\,\mathrm{m\,s^{-1}}$). A $3.7\,\mathrm{m\,s^{-1}}$). Assuming a constant emission, a wind speed higher by a factor of $\sim\!1.6$ roughly 2 means a decrease in the column enhancements by a factor of $\sim\!1.6$, which reduced roughly 2, which would significantly reduce the visibility of the plume in the retrieval results by diluting larger parts of the plume faster below the background noise.

In Fig. 8, a new interesting feature was observable at the source. There, we observed a double plume structure that was especially prominent during the first overpass. Comparison with the RGB image revealed, that one part seemed to originate from the vent, while the other part seemed to originate from the top of the shadow of the vent. The vent released the plume

several meters above the surface. Because the plume was very narrow near the source, the sunlight only passed the plume either before or after hitting the ground. As those two light paths were attributed to different ground scenes, the absorption and therefore the apparent $CH_4$ enhancements were visible at two locations leading to the double plume structure. Further down the plume, atmospheric mixing took place and the plume widened. Then, the sunlight passed through the plume both before and after hitting the ground, and the double plume structure vanished. A simple geometric consideration of the distance between the two plume structures ($\sim 50\,\mathrm{m}$), the solar and instrument zenith  angles ($\sim 42\,°$ and $\sim 2.5\,°$), and the vent height  calculated from the shadow of the structure ($\sim 55\,\mathrm{m}$) and the solar zenith angle supported this hypothesis.

In Fig. 9, two plumes originating from well pads are shown. Both extended linearly from their source and were visible over approximately $100\,\mathrm{m}$. While the first plume (Fig. 9, b) originated from a cold vent (similar to Fig. 8), the emitting structure for the plume in Fig. 9 (d) could not be identified from the RGB images. It seemed, however, that the source was located near the surface. This would also explain the large deviation of the plume direction from the wind direction aquired from the ERA-5 model data, since the nearby forests could have significantly altered the wind direction near the surface.

Additionally, the plume detected over a coal mine ventilation shaft during the Four Corners measurement campaign in 2015 is shown in Fig. 10. The plume was nearly one kilometer long, with a straight profile except for a small diversion at the tip that suggested a change in wind direction.

**4.2   Comparison of WFM-DOAS retrieval results with IMAP-DOAS and MF results**

To assess the performance of the WFM-DOAS retrieval with respect to the IMAP-DOAS and MF retrieval results, a subset of the data was compared quantitatively. We focused here on the coal mine ventilation shaft plume for all three retrievals, and additionally compared the WFM-DOAS results to MF results over the plumes P1 - P4. As IMAP-DOAS and the MF retrieve $CH_4$ enhancements below the aircraft in $\mathrm{ppm} \cdot \mathrm{m}$ ($CH_{4,\mathrm{enh,ppm\,m}}$), we converted these measurements to enhancements in $\mathrm{molec\,cm^{-2}}$ ($CH_{4,\mathrm{enh,MF}}$) via the following equation:

$$CH_{4,\mathrm{enh,MF/IMAP}} = \frac{CH_{4,\mathrm{enh,ppm\,m}}}{h_{\mathrm{airc}}} \cdot \mathrm{subcol_{tot}} \cdot 10^{-6}, \tag{3}$$

where $h_{\mathrm{airc}}$ is the distance between instrument and ground in nadir direction, and $\mathrm{subcol_{tot}}$ is the number of dry molecules per square centimeter between aircraft and ground. The sub column below the aircraft was calculated on the basis of the profiles used for the WFM-DOAS retrieval.

We compared the retrieval results in two different ways, focusing on retrieval scatter and retrieved enhancements. First, for the retrieval scatter, we calculated the standard error (1-sigma) of retrieved enhancements over areas near the plumes P1 - P5. This is summarized in Table 5. For all plume regions except P5 the retrieval scatter is lower for WFM-DOAS than for the MF results, which indicate less retrieval noise. For P5, where all three retrieval results were available, the background noise is very similar for all retrieval methods.

Second, to evaluate the agreement of the retrieval results, we additionally produced scatter plots of the central plume area. We added the 1:1 line and a linear fit to the scatter plots. For the coal mine ventilation shaft plume all three retrievals agreed well

**Table 5.** Comparison of background noise of the retrievals based on the standard deviation of retrieval results near the plumes. As we only have IMAP-DOAS results for one plume, the other fields are left empty.

| | P1 (molec cm$^{-2}$) | P2 (molec cm$^{-2}$) | P3 (molec cm$^{-2}$) | P4 (molec cm$^{-2}$) | P5 (molec cm$^{-2}$) |
|---|---|---|---|---|---|
| WFM-DOAS | $\pm 1.3 \cdot 10^{17}$ | $\pm 8.3 \cdot 10^{17}$ | $\pm 7.8 \cdot 10^{17}$ | $\pm 1.8 \cdot 10^{18}$ | $\pm 5.2 \cdot 10^{17}$ |
| MF | $\pm 2.0 \cdot 10^{18}$ | $\pm 2.1 \cdot 10^{18}$ | $\pm 1.7 \cdot 10^{18}$ | $\pm 2.4 \cdot 10^{18}$ | $\pm 4.1 \cdot 10^{17}$ |
| IMAP-DOAS | | | | | $\pm 5.5 \cdot 10^{17}$ |

in the retrieved enhancements, as can be seen in Fig. 11. Compared to the IMAP-DOAS retrieval, WFM-DOAS did retrieve slightly lower enhancements, while it did estimate slightly higher enhancements than MF.

Additionally, we compared the WFM-DOAS and MF results for the plumes P1 - P4 (see Fig. D1). For P3 especially the higher enhancements located inside the plume agree well. However, for the other plumes there is quite some mismatch between
5   MF results and WFM-DOAS results. The cause for the disagreement is currently under investigation and might be related to the use of a standard gas target for the matched filter rather than calculating a target specifically parameterized for a given scene.

**4.3    Flux and uncertainty estimation based on cross sectional flux method**

[revised manuscript text omitted]

**Appendix A: Comparison of weather station wind data with ERA5 50 m height average data**

We compared the average wind speed over the lowest $50\,\mathrm{m}$ height average of ERA5 data to the hourly mean wind speed data

25   obtained from weather stations near the sources. For P1 and P2, we used wind speed data obtained at the Firebag weather station (Wood Buffalo Environmental Association, 2020) which was located slightly below $20\,\mathrm{km}$ away from the plumes. For P3 and P4, we used wind speed data from the weather stations "Sundre A" and "Patricia AGCM"[1], located $5\,\mathrm{km}$ and $17\,\mathrm{km}$ away from the source. For P5, we used wind speed data from the Four Corners Regional Airport weather station from the MesoWest network (Horel et al., 01 Feb. 2002) $\sim 15\,\mathrm{km}$ away from the plume. A comparison between those ground stations

30   and the average wind speed over the lowest $50\,\mathrm{m}$ of ERA5 data is shown in Fig. A1.

  [1]Data provided by Alberta Agriculture and Forestry, Alberta Climate Information Service (ACIS) https://acis.alberta.ca (retrieved in Dec. 2020)

As is clearly visible, the ERA5 data significantly deviated from the wind speeds present at the weather stations. The mean deviation was $+2.5\,\mathrm{m\,s^{-1}}$, with single differences up to $+3.4\,\mathrm{m\,s^{-1}}$. Consequently, we used the weather station wind data for further analysis in the flux inversion.

**Appendix B: Striping Effect**

[revised manuscript text omitted]

**Flight line ang20150422t164506 - Source P5**

[Figure]

**Figure 10.** Similar to Fig. 8, but for the exhaust plume of a coal mine ventilation shaft in the Four Corners region.

[Figure]

**Figure 11.** Scatter plot comparison between IMAP-DOAS and WFM-DOAS (a) and MF and IMAP-DOAS (b) respectively for the coal mine ventilation shaft plume. For the red line only enhancements above the background noise stated in Table 5 were analyzed, which should be largely dominated by the plume.

[Figure]

**Figure A1.** Comparison between wind speed averaged over the lowest 50 m above ground for ERA5 data and collocated wind station data. ERA5 data significantly underestimated the wind speed present at a given time.

[Figure]

**Figure C1.**  Same as Fig. 9, but for methane enhancements after application of the proxy resulting from bituminous coal extraction.  The upper two measurements  were taken within ∼ 10 minutes, and show the  same brim. The dashed line is plotted on the same location in both images. While in (b) a plume emanating from the brim is visible, in (d) strong accumulations near the brim are visible. In (f) the plume is more diffuse and the highest enhancements are  visible near a brim. However, it is not as clear as in (b) and (d), if it really originates from the brim or might be an artifact.

[Figure]

**Figure C2.** Similar to Fig. C1, but for methane enhancements from oil/gas infrastructure. In (a) and (b), the emissions originate from a well pad in a forest. Due to the low radiance over the trees, only the well pad itself passes the quality filters. The enhancements seem accumulated and no clear wind direction is visible. In (c) and (d), a facility located at a bitumen extraction site is shown and meanders through the facility.

[Figure]

**Figure C3.** Similar to Fig. C1, but for two overpasses over cold vents at a bitumen extraction site. The plumes are only faint, especially for the second overpass (d). In the results a small striping effect is still visible. The destriping reduces the effect, but is not able to totally eliminate it.

**Appendix D: Additional comparison between MF and WFM-DOAS**

In addition to the comparison of IMAP-DOAS and MF for the same plume, we compared MF and WFM-DOAS retrieval results for subregions zoomed in on the plumes P1 - P4. For the scene P3 the retrievals seem to agree quite well. Especially the higher enhancements correlated with the plume lay near the 1:1 line. However, for the plumes P1, P2 and P4, there is a bigger discrepancy between MF and WFM-DOAS, which can not be explained by the larger scatter alone. The cause for this is currently under investigation.

[Figure]

**Figure D1.** Similar to Fig. 11, but for the plume regions P1 - P4.

*Author contributions.* JB contributed to the study design, adapted the retrieval, analyzed the data and wrote the manuscript, KG and HB initialized the study, KG and SK contributed to the retrieval adaption, study design and paper draft, AKT, DRT and CF helped in data set selection and with the handling of the data, RD and CM designed the flight plans, organized and led the flights, HB and JPB supervised the study, contributing to the scientific objectives and interpretation of results. All authors contributed to the final manuscript.

5    *Competing interests.* The authors declare that they have no conflict of interest.

*Acknowledgements.* This work was funded in parts by the BMBF project AIRSPACE MAMAP2D (FKZ01LK1701B). We would also like to acknowledge the contributions of the AVIRIS flight and instrument teams. The AVIRIS-NG data were collected as part of the Arctic Boreal Vulnerability Experiment (ABoVE), a NASA Terrestrial Ecology program, and other methane studies supported by NASA's Earth Science Division. A portion of this work was performed at the Jet Propulsion Laboratory, California Institute of Technology, under contract

10   with NASA. The hourly resolved wind data for P3 and P4 were provided provided by Alberta Agriculture and Forestry, Alberta Climate Information Service (ACIS), https://acis.alberta.ca for stations Patricia AGCM and Sundre A (Data retrieved on Dec, 2020)